# Diffusion Feedback Helps CLIP See Better

**Wenxuan Wang**[1,2,3]* **Quan Sun**[3]* **Fan Zhang**[3] **Yepeng Tang**[4] **Jing Liu**[1,2] **Xinlong Wang**[3]†

[1] Institute of Automation, Chinese Academy of Sciences
[2] School of Artificial Intelligence, University of Chinese Academy of Sciences
[3] Beijing Academy of Artificial Intelligence
[4] Institute of Information Science, Beijing Jiaotong University

Project Page: *https://rubics-xuan.github.io/DIVA/*

## Abstract

Contrastive Language-Image Pre-training (CLIP), which excels at abstracting open-world representations across domains and modalities, has become a foundation for a variety of vision and multimodal tasks. However, recent studies reveal that CLIP has severe visual shortcomings, such as which can hardly distinguish orientation, quantity, color, structure, *etc*. These visual shortcomings also limit the perception capabilities of multimodal large language models (MLLMs) built on CLIP. The main reason could be that the image-text pairs used to train CLIP are inherently biased, due to the lack of the distinctiveness of the text and the diversity of images. In this work, we present a simple post-training approach for CLIP models, which largely overcomes its visual shortcomings via a self-supervised diffusion process. We introduce **DIVA**, which uses the **DI**ffusion model as a **V**isual **A**ssistant for CLIP. Specifically, **DIVA** leverages generative feedback from text-to-image diffusion models to optimize CLIP representations, with only images (without corresponding text). We demonstrate that **DIVA** improves CLIP's performance on the challenging MMVP-VLM benchmark which assesses fine-grained visual abilities to a large extent (*e.g.*, 3-7% ↑), and enhances the performance of MLLMs and vision models on multimodal understanding and segmentation tasks. Extensive evaluation on 29 image classification and retrieval benchmarks confirms that our framework preserves CLIP's strong zero-shot capabilities. The code is publicly available at https://github.com/baaivision/DIVA.

## 1 Introduction

Contrastive language-image pre-training (CLIP) (Radford et al., 2021b) has been widely applied to various multimodal understanding and generation tasks, including open-domain image classification (Sun et al., 2024d; Zhang et al., 2022; Zhu et al., 2023), text-to-image retrieval (Luo et al., 2023; Baldrati et al., 2022; Sain et al., 2023), visual grounding (Wang et al., 2022; Yu et al., 2023; Wang et al., 2024b;a), and text-to-image generation (Frans et al., 2022; Bar-Tal et al., 2022; Rombach et al., 2022a; Crowson et al., 2022; Ramesh et al., 2022; Vinker et al., 2022). This widespread application is due to CLIP's excellent visual representation ability, learned from large-scale data. Thus, enhancing CLIP's representation and capabilities is crucial for advancing downstream tasks.

Since the introduction of CLIP (Radford et al., 2021b), numerous subsequent studies on CLIP models have emerged in recent years. These studies have utilized training techniques such as pre-training (Sun et al., 2023; 2024b; Fang et al., 2023; Xu et al., 2023a; Zhai et al., 2023; Shi et al., 2024) and fine-tuning (Wei et al., 2023b; Zhang et al., 2024) CLIP models, achieving improved performance and unlocking new abilities. However, these approaches still suffer from unavoidable limitations, as they heavily rely on image-text data pairs and cannot work on image-only data.

As noted by recent works (Kim et al., 2023; Zeng et al., 2021; Zhang et al., 2024; Tong et al., 2024b;a), despite its excellent zero-shot performance, CLIP suffers from certain perceptual understanding limitations due to the contrastive learning paradigm and the noisy image-text pairs used in training.

---

*Equal contribution. † Correspondence to *wangxinlong@baai.ac.cn*.

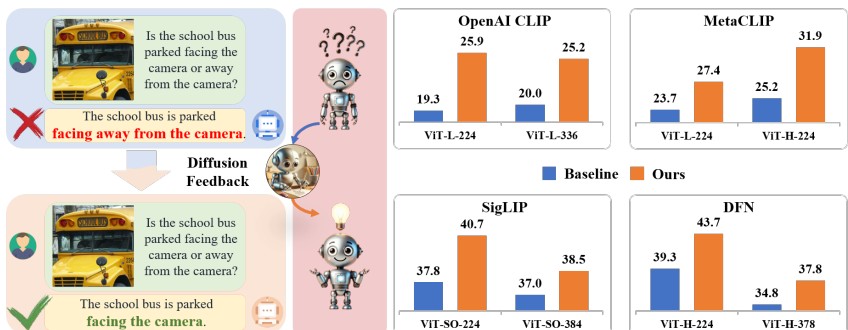

Figure 1: **Left:** The existing CLIP models mostly suffer from the inability to distinguish visual details. After enhancing the visual capabilities with our **DIVA**, the sensitivity of CLIP to visual details has greatly improved. **Right:** Our proposed **DIVA** consistently boosts the performance of various CLIP models (Radford et al., 2021b; Fang et al., 2023; Xu et al., 2023a; Zhai et al., 2023) on MMVP-VLM benchmark that evaluates the visual capabilities of vision-language models.

These limitations include an inability to accurately comprehend long texts and to perceive fine-grained differences in similar images. While some studies have attempted to address the text comprehension issue (Zhang et al., 2024), research on improving CLIP's fine-grained visual perception remains underexplored. As studied in (Kim et al., 2023), the perceptual shortcomings of CLIP are closely tied to its visual encoder's inability to grasp fine-grained visual details, which is crucial for multimodal models. The lack of this capability in CLIP directly affects the performance of vision and multimodal models that use CLIP as a vision encoder (Tong et al., 2024b;a). As demonstrated in Fig. 1, existing CLIP models mostly struggle to distinguish subtle visual differences between similar images, highlighting the fundamental limitation in fine-grained visual perception. To mitigate this visual perception shortcoming of CLIP, lots of recent works such as (Tong et al., 2024b; Kar et al., 2025; Zong et al., 2024) have attempted to improve multimodal large language models (MLLMs) by incorporating extra vision-only self-supervised backbones, such as DINOv2 (Caron et al., 2021), to enhance visual perception capability. However, these approaches still do not fundamentally address CLIP's visual perception limitations, and often introduce significant computational costs.

Thus, in this work, we aim to fundamentally address CLIP's inability to distinguish fine-grained visual details via SSL paradigm. Considering that diffusion models can naturally generate detailed and realistic images, text-to-image generation inherently requires the diffusion models to grasp fine-grained visual representations (Wei et al., 2023a). Motivated by this, we propose utilizing the generative feedback from diffusion models to enhance CLIP's visual perception capabilities, which is solely driven by image data with manageable training cost. By conditioning diffusion models with CLIP's densely recapped visual features and applying reconstruction loss for CLIP's representation optimization, we leverage the **DI**ffusion model as a **V**isual **A**ssistant for CLIP, thus the name of our approach, **DIVA**. Our results highlight that **DIVA** greatly enhances CLIP's performance on MMVP-VLM benchmark measuring visual abilities of vision-language (V-L) models, and improves MLLMs and vision models on multimodal and vision understanding tasks. Besides, **DIVA** maintains CLIP's excellent zero-shot performance on 29 image classification and retrieval benchmarks. The key advantage of **DIVA** lies in its ability to enhance CLIP's visual perception capabilities through a resource-efficient post-training process, using only image data in a self-supervised manner, without the need for expensive image-text pair data collection or introducing additional visual encoders.

Our main contributions can be summarized as follows:

- Concentrating on overcoming CLIP's visual shortcomings in perceiving fine-grained details, we present the first work to exploit the potential of leveraging generative feedback from text-to-image diffusion models to optimize CLIP model's discriminative representations.
- We propose a simple self-supervised framework **DIVA** for CLIP's representation optimization. Coupled with our visual dense recap scheme, **DIVA** conditions diffusion models with dense visual features from CLIP and incorporates image reconstruction loss for optimization.
- Our **DIVA** greatly boosts CLIP's visual perception capability and improves its performance on MMVP-VLM benchmark, further enhancing MLLMs and vision models on multimodal and visual understanding tasks. Meanwhile, our results on 29 image classification and retrieval benchmarks show that **DIVA** maintains CLIP's original excellent zero-shot performance.

## 2 RELATED WORK

**CLIP Models & MLLMs.** The introduction of CLIP (Radford et al., 2021b) has significantly advanced multimodal learning. Since its debut, a series of CLIP models have emerged (Sun et al., 2023; Fang et al., 2023; Xu et al., 2023a; Zhai et al., 2023), enhancing performance and unlocking new capabilities through improved pre-training techniques and model architectures. On this basis, CLIP has been widely adopted as a foundation model, serving as a backbone for various applications such as image segmentation (Li et al., 2022a; Xu et al., 2022; Shan et al., 2024; Xu et al., 2023c; Liang et al., 2023; Zhou et al., 2023), object detection (Gu et al., 2021; Li et al., 2022b; Subramanian et al., 2022) and video understanding (Bose et al., 2023; Lin et al., 2022; Castro & Heilbron, 2022; Xu et al., 2021; Rasheed et al., 2023; Tang et al., 2024). Its ability to align language and vision has led to superior results on these tasks compared to traditional methods. Moreover, CLIP has driven the development of MLLMs (Liu et al., 2024b;a; Sun et al., 2024c;a). Combining strong visual understanding with advanced large language models facilitates more sophisticated interactions between vision and language. Recent works have highlighted inherent visual flaws in the CLIP models and MLLMs using CLIP as the visual encoder (Tong et al., 2024b;a). To address this, some research has incorporated multiple vision encoders to achieve more precise and comprehensive visual perception (Kar et al., 2024; Jiang et al., 2023; Tong et al., 2024b). However, this approach increases computational costs and memory usage. There has been no research directly enhancing CLIP's visual perception capabilities to better serve MLLMs. Thus, the main focus of our work is to fundamentally overcome CLIP's visual perception shortcomings, directly benefiting both vision models and multimodal MLLMs that use CLIP as a backbone.

**Diffusion Models for Representation Learning.** Diffusion models (Ho et al., 2020; Song et al., 2020) have made remarkable progress in various generative tasks, such as image generation (Rombach et al., 2022b; Saharia et al., 2022; Betker et al., 2023; Zheng et al., 2024), video generation (Singer et al., 2022; Blattmann et al., 2023; Junhao Zhang et al., 2023; Ho et al., 2022), editing (Meng et al., 2021; Hu et al., 2024; Mou et al., 2023), etc. Apart from research above, there are also many works focus on employing diffusion models for representation learning. Some of works leverage the intermediate activation of pre-trained diffusion models for different downstream tasks, including classification (Xiang et al., 2023), semantic segmentation (Baranchuk et al., 2021), panoptic segmentation (Xu et al., 2023b), depth estimation (Zhao et al., 2023), etc. Other works (Hudson et al., 2024; Pan et al., 2023) train their own diffusion models coupled with meticulously devised modules to further boosting the representation capabilities. Besides, Diffusion-TTA (Prabhudesai et al., 2023) aims to adapt pre-trained vision encoders to samples in testing set using feedback from a diffusion model. Additionally, some methods (Guo et al., 2024; Trabucco et al., 2023; Tian et al., 2024; Azizi et al., 2023) utilize diffusion models to generate synthetic data, which is then adopted to enhance the representation capabilities of corresponding models. In contrast, in our work, we mainly focus on exploring the potential of enhancing the original discriminative representations of CLIP models by directly utilizing generative feedback from the diffusion models. Additionally, we aim to leverage the diffusion models to break free from the constraints of paired image-text data and construct a self-supervised framework to improve CLIP's visual perception capabilities.

## 3 ENHANCING CLIP'S REPRESENTATIONS VIA DIFFUSION FEEDBACK

In this section, we present our **DIVA**, an effective framework for boosting CLIP's visual perception capabilities with a pre-trained conditional diffusion model. We first discuss CLIP's visual deficiencies in perceiving details and generative diffusion models as preliminaries in Sec. 3.1.1 and Sec. 3.1.2 respectively. Then the overall architecture of **DIVA** is illustrated in Sec. 3.2, followed by our well-designed visual dense recap strategy for better unleashing the power of **DIVA** in Sec. 3.3.

### 3.1 PRELIMINARIES

#### 3.1.1 CLIP'S VISUAL DEFICIENCIES

Thanks to the excellent representations learned through pre-training on massive data, CLIP (Radford et al., 2021b) exhibits outstanding generalization capabilities and is widely applied in the V-L understanding domain. However, CLIP is not without its flaws. As highlighted in the study (Tong et al., 2024b), CLIP struggles to distinguish detailed differences between two images that are obviously distinct to human observers. This deficiency mainly stems from two aspects: 1) **Training**

**Paradigm**: The original contrastive learning strategy of CLIP aims to minimize the distance between positive pairs and maximize the distance between negative pairs of visual class tokens and textual semantics, resulting in visual perception bias that mainly focuses on high-level semantic information while overlooking visual details such as orientation, quantity, color, and structure. Consequently, CLIP sometimes encodes visually different images into similar embeddings, making it difficult to differentiate these images' subtle variations. 2) **Data Format**: The text in the image-text pairs used to train CLIP is limited in length. As pointed out by (Zhang et al., 2024), although the length of the text token is restricted to 77, CLIP's actual effective text length is less than 20. Therefore, the textual data in these image-text pairs inherently lacks descriptions of the visual details in the corresponding positive sample images. This fundamental limitation of the training data also leads to CLIP's inability to adequately perceive visual detail information.

### 3.1.2 GENERATIVE DIFFUSION MODELS

Generative diffusion models have proven to be highly effective in modeling high-dimensional data and have become the de facto approach for visual generation tasks (Podell et al., 2023; Yang et al., 2024). They possess strong capabilities in capturing intricate visual details. A diffusion model reconstructs an image by gradually removing noise added during the forward process. Formally, given an image sample $x_0$ drawn from an underlying probability distribution $p(x)$, a forward diffusion process defines a Markov chain to gradually add random Gaussian noise $\epsilon_{\mathbf{t}} \in \mathcal{N}(\mathbf{0}, \mathbf{I})$ to the original sample $\mathbf{x_0}$:

$$\mathbf{x_t} = \sqrt{1 - \beta_t}\mathbf{x_{t-1}} + \sqrt{\beta_t}\epsilon_{\mathbf{t}}, \quad t = 1, \ldots, T \tag{1}$$

Here, $T$ denotes the number of diffusion steps, and $\beta_t \in (0, 1)$ is a predefined time-dependent variance schedule. As the $T$ becomes large enough, $\mathbf{x_T}$ is close to $\mathcal{N}(\mathbf{0}, \mathbf{I})$. The transition equation can be reformulated as follows by leveraging the additive property of Gaussian distribution:

$$\mathbf{x_t} = \sqrt{\bar{\alpha}_t}\mathbf{x_0} + \sqrt{1 - \bar{\alpha}_t}\epsilon, \quad t = 1, \ldots, T \tag{2}$$

in which $\alpha_t = 1 - \beta_t$ and $\bar{\alpha}_t = \prod_{i=1}^{t} \alpha_i$. On this basis, the image sample $x_0$ can be iteratively generated from a random noise $\mathbf{x_T} \sim \mathcal{N}(\mathbf{0}, \mathbf{I})$ by reversing the forward diffusion process:

$$\mathbf{x_{t-1}} = \frac{1}{\sqrt{\alpha_t}}(\mathbf{x_t} - \frac{1 - \alpha_t}{\sqrt{1 - \bar{\alpha}_t}}\epsilon_\phi(\mathbf{x_t}, t)) + \sigma_t\epsilon, \quad t = T, \ldots, 1 \tag{3}$$

$\epsilon_\phi$ is a denoising neural network trained to predict $\epsilon$ in the forward diffusion process and $\sigma_t$ is the posterior noise variance. A commonly used training objective for a diffusion model $\epsilon_\phi$ is:

$$L(\phi) = \mathbb{E}_{t,\mathbf{x_0},\epsilon} \left[ \|\epsilon - \epsilon_\phi(\sqrt{\bar{\alpha}_t}\mathbf{x_0} + \sqrt{1 - \bar{\alpha}_t}\epsilon, t)\|^2 \right] \tag{4}$$

Besides, the diffusion models can be easily extended to conditional generation by incorporating a condition $\mathbf{c}$ into $\epsilon_\phi$, in which $\mathbf{c}$ can be a class label, a text prompt, etc. Thus, the training objective should be modified to:

$$L(\phi) = \mathbb{E}_{t,\mathbf{x_0},\epsilon,\mathbf{c}} \left[ \|\epsilon - \epsilon_\phi(\sqrt{\bar{\alpha}_t}\mathbf{x_0} + \sqrt{1 - \bar{\alpha}_t}\epsilon, t, \mathbf{c})\|^2 \right] \tag{5}$$

In practice, modulation layers or cross-attention layers can be employed to integrate conditions into denoising neural network, guiding the denosing process.

### 3.2 OVERALL STRUCTURE OF **DIVA**

As illustrated in Fig. 2, **DIVA** mainly consists of two parts: the CLIP model to be enhanced in terms of visual perception capabilities, and the pre-trained text-to-image diffusion model providing generative feedback. Taking an original image as input, the CLIP model encodes the corresponding visual features, which will be combined with the empty text's embeddings (*i.e.*, [BOS] & [EOS]) from the diffusion model's text encoder for diffusion's condition. Given the image with added noise, the diffusion model attempts to predict the noise added from the previous step to the current step with the aforementioned condition. This process needs to be repeated N times because, for each image, we will randomly select N states with a uniform distribution from the total steps (*e.g.*, 0~1000 steps) of the diffusion model for optimization. The corresponding loss function can be represented as Equation 5. Keeping all parts' weights except the CLIP visual encoder frozen, the training objective is simply to minimize the reconstruction loss (*i.e.*, generative guidance). In this manner, by constraining

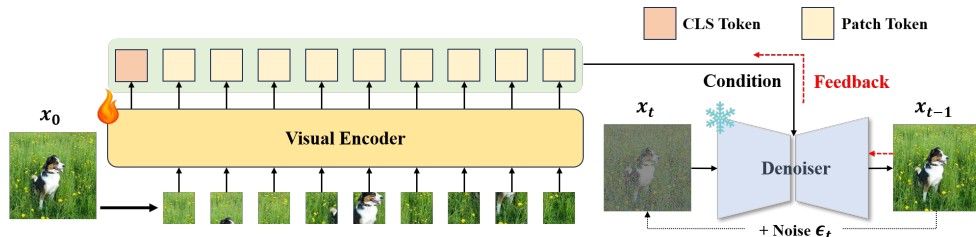

Figure 2: **Overall architecture of our DIVA.** Given an image $x_0$, the CLIP model $\theta$ encodes the visual features as main part of condition **c**, then the generative diffusion model $\phi$ predicts the added noise $\epsilon$ taking the noisy image $x_t$ and condition **c** as input. We optimize the CLIP's representation by maximizing the image likelihood with the diffusion loss via generative feedback.

the diffusion model to more accurately predict the added noise, the CLIP's original semantic-rich discriminative representations will be gradually optimized into representations with more visual details through diffusion feedback. On this basis, results in Sec. 4.4 demonstrate that our **DIVA** does not greatly damage the zero-shot performance of the original CLIP models. The pseudo code of the specific enhancement process can be found at Algorithm 1 in Appendix.

### 3.3 DIFFUSION'S VISUAL CONDITION DESIGN

In our **DIVA**, the diffusion model's condition design is pivotal, as it sets the upper limit for enhancing CLIP's visual capabilities. Since our motivation for designing DIVA is to enhance CLIP's ability to capture fine-grained details through a resource-efficient, purely image-driven self-supervised framework for post-training, we do not introduce aligned textual information as the diffusion condition. However, bridging the gap between pure visual features and text embeddings for the diffusion process is crucial. To address this, we first pass the visual features through a vision-language projection layer to map the high-level representations. Additionally, we incorporate the embedding of an empty text as part of the diffusion model's condition to further narrow this gap. In this way, our condition design resembles an empty text shell, but the actual content inside is akin to a "visual sentence" obtained via vision-to-text mapping. This design more seamlessly fits the diffusion model's conditional input.

**Visual Dense Recap Scheme.** Apart from the aforementioned empty text's embeddings from diffusion model's text encoder, we introduce a simple yet effective strategy called visual dense recap for the designing of visual condition part. Unlike detailed re-captioning of an image's caption in natural language, our approach performs re-captioning at the level of visual richness by incorporating features from local patch tokens along with the class token into the condition. When only the class token is present, CLIP's visual features primarily contain strong semantic information, which is insufficient for reconstructing the original image. Consequently, the reconstruction task becomes challenging due to the lack of adequate information, and CLIP cannot learn significantly enhanced representations. By incorporating local patch features, the auxiliary function of the condition is significantly enhanced, allowing the generative feedback to effectively improve CLIP's visual perception capabilities. We conduct ablation studies in Sec. 4.5 to demonstrate the efficacy of visual dense recap.

**Visual Recap Density.** Although the visual dense recap scheme appears straightforward, the density of the recap is crucial. If the density is too high (*i.e.*, introducing too many local tokens), the richness of the condition information approaches its maximum, greatly reducing the difficulty of reconstruction task. This results in CLIP's representation requiring minimal optimization to easily complete the reconstruction, limiting the upper bound of CLIP's optimized capabilities. Conversely, if the recap density is too low (*i.e.*, retaining only class token or introducing few local tokens), CLIP's optimization process will struggle with the high difficulty of reconstruction, failing to adequately learn the expected detailed visual representations. This intuitive point is confirmed in Sec. 4.5.

In fact, our **DIVA** can enhance different CLIP models' fine-grained perception abilities through a general and simple condition design principle, *e.g.*, introducing all the local patch tokens along with the class token as the visual condition for diffusion model. However, we empirically believe that each baseline's requirement for visual density is actually different, which is also supported by our preliminary experiments. This is because different CLIP models are trained on vastly different data, leading to significant variations in their learned representations. Therefore, their visual density for various CLIP models needs to be specifically adjusted to maximize the learning of corresponding detailed representations through generative feedback. Specifically, ensuring the

visual class token is always present in the condition, we introduce randomly selected local token features with approximately 15% and 30% probabilities for OpenAI CLIP (Radford et al., 2021b) at 224 and 336 resolutions. For the SigLIP ViT-SO-14 (Zhai et al., 2023) at 224 and 384 image sizes, we incorporate local token features obtained through 1D average pooling with local window sizes of 6 and 10, respectively. Except that introducing 50% randomly selected patch tokens into the condition for DFN ViT-H-14/378, for the remaining baselines (Fang et al., 2023; Xu et al., 2023a), we include all local token features for the condition design. Apart from DFN ViT-H-14/224 and SigLIP ViT-SO-14/224&384 (Zhai et al., 2023) only using visual class token, all other models incorporate local features consistent with the training stage conditions during inference, combining them with class token to fully leverage the detailed representations captured by the enhanced CLIP.

## 4 EXPERIMENTAL RESULTS

To evaluate the effectiveness of our **DIVA** and demonstrate its potential to enhance CLIP representations, comprehensive experiments are conducted on multimodal understanding and visual perception tasks, which will be elaborated in the followings.

### 4.1 IMPLEMENTATION DETAILS

**DIVA** is trained on 8 NVIDIA-A100 80GB GPUs with a batch size of 640. We adopt Stochastic Gradient Descent (SGD) optimizer with a learning rate of 1e-4 and momentum of 0.9 to refine CLIPs' representations via generative feedback. We only optimize the CLIP models with relatively high-quality Conceptual-3M dataset (Sharma et al., 2018) for 4600 steps (*i.e.*, nearly 1 epoch) during training, which can already boost CLIP's performance in a training-efficient manner. For all experiments, we adjust the parameters of the discriminative CLIP vision encoders and keep the pre-trained diffusion models frozen through the training process. Besides, the specific dataset and evaluation details about the MMVP and MMVP-VLM benchmark can be found in Appendix.

Regarding the diffusion sampling step N, our choice is made by balancing training cost and model performance gains. Specifically, we begin with the initial state of N=1 to enhance the representation quality of CLIP vision encoder. As N increases, the training cost rises significantly. When increasing N from 1 to 2 (meaning that each image undergo diffusion sampling twice to provide two rounds of generative feedback for CLIP model optimization), performance gains are observed. However, further increasing N beyond 2 not only greatly escalate training costs but also do not yield additional benefits for the CLIP model's representation learning. Therefore, N=2 is selected as the optimal sampling step to consistently improve performance across various baselines.

### 4.2 FINE-GRAINED VISUAL PERCEPTION EVALUATION

To validate that our **DIVA** can effectively mitigate the inherent visual capability deficiencies of CLIP models, we first conduct experiment on various existing CLIP models (Radford et al., 2021b; Fang et al., 2023; Xu et al., 2023a; Zhai et al., 2023). Despite the variations in image resolution, model size, training data and methodology among these CLIP models, our method consistently enhances their performances on the MMVP-VLM benchmark. As presented in Table 1, our framework achieves the best performance improvements (*i.e.*, ↑4-7%) on OpenAI ViT-L-14 and MetaCLIP ViT-H-14, and even on current best-performing DFN ViT-H-14 our framework realizes a performance gain of nearly 3-5%. This fully demonstrates that **DIVA** is both general and effective in enhancing the fine-grained visual perception capabilities of CLIP models. Notably, through the generative guidance provided by our self-supervised framework that is free from image-text constraints, the perceptual abilities of CLIP models on almost all visual patterns have the potential to be enhanced.

### 4.3 BACKBONE ENHANCEMENT PERFORMANCE EVALUATION

Next, with the help of our **DIVA**, we further evaluate the performance gains brought by the enhanced CLIP backbones for multimodal understanding and visual perception tasks.
**Enhanced Vision Backbone for MLLMs.** Firstly, we adopt LLaVA-1.5 (Liu et al., 2024a) as the baseline framework to explore the potential of improved visual encoders in MLLM. LLaVA employs a pre-trained CLIP vision encoder and trains a projector to semantically align visual tokens with textual tokens from large language model (LLM). To ensure fair comparisons, we train our model with the same setting in LLaVA and evaluate model performance on various multimodal understanding benchmarks (*i.e.*, MMVP (Tong et al., 2024b), POPE (Li et al., 2023), MME-Perception (Fu et al.,

Table 1: **Performance of CLIP based models on various visual patterns of MMVP-VLM benchmark.** Our framework greatly overcomes CLIP's original shortcomings in terms of perceiving visual details. Symbols for visual patterns as (Tong et al., 2024b) are inherited: ⊘: Orientation and Direction, ℚ: Presence of Specific Features, ⟳: State and Condition, ↕: Quantity and Count, ⚲: Positional and Relational Context, 🎨: Color and Appearance, ⚙: Structural and Physical Characteristics, **A**: Texts, 📷: Viewpoint and Perspective.

| Method | Ours | Image Size | Params (M) | ⊘ | ℚ | ⟳ | ↕ | ⚲ | 🎨 | ⚙ | A | 📷 | Average |
|---|---|---|---|---|---|---|---|---|---|---|---|---|---|
| OpenAI ViT-L-14 | | $224^2$ | 427.6 | 13.3 | 13.3 | 20.0 | 20.0 | 13.3 | 53.3 | 20.0 | 6.7 | 13.3 | 19.3 |
| OpenAI ViT-L-14 | ✔ | $224^2$ | 427.6 | 13.3 | 20.0 | 40.0 | 6.7 | 20.0 | 53.3 | 46.7 | 20.0 | 13.3 | **25.9** (+6.6) |
| OpenAI ViT-L-14 | | $336^2$ | 427.9 | 0.0 | 20.0 | 40.0 | 20.0 | 6.7 | 20.0 | 33.3 | 6.7 | 33.3 | 20.0 |
| OpenAI ViT-L-14 | ✔ | $336^2$ | 427.9 | 26.7 | 20.0 | 33.3 | 13.3 | 13.3 | 46.7 | 26.7 | 6.7 | 40.0 | **25.2** (+5.2) |
| MetaCLIP ViT-L-14 | | $224^2$ | 427.6 | 13.3 | 6.7 | 66.7 | 6.7 | 33.3 | 46.7 | 20.0 | 6.7 | 13.3 | 23.7 |
| MetaCLIP ViT-L-14 | ✔ | $224^2$ | 427.6 | 6.7 | 6.7 | 60.0 | 0.0 | 26.7 | 66.7 | 20.0 | 20.0 | 40.0 | **27.4** (+3.7) |
| MetaCLIP ViT-H-14 | | $224^2$ | 986.1 | 6.7 | 13.3 | 60.0 | 13.3 | 6.7 | 53.3 | 26.7 | 13.3 | 33.3 | 25.2 |
| MetaCLIP ViT-H-14 | ✔ | $224^2$ | 986.1 | 13.3 | 20.0 | 53.3 | 33.3 | 13.3 | 66.7 | 33.3 | 13.3 | 40.0 | **31.9** (+6.7) |
| SigLIP ViT-SO-14 | | $224^2$ | 877.4 | 26.7 | 20.0 | 53.3 | 40.0 | 20.0 | 66.7 | 40.0 | 20.0 | 53.3 | 37.8 |
| SigLIP ViT-SO-14 | ✔ | $224^2$ | 877.4 | 13.3 | 26.7 | 60.0 | 46.7 | 13.3 | 73.3 | 53.3 | 26.7 | 53.3 | **40.7** (+2.9) |
| SigLIP ViT-SO-14 | | $384^2$ | 878.0 | 20.0 | 26.7 | 60.0 | 33.3 | 13.3 | 66.7 | 33.3 | 26.7 | 53.3 | 37.0 |
| SigLIP ViT-SO-14 | ✔ | $384^2$ | 878.0 | 26.7 | 33.3 | 53.3 | 26.7 | 13.3 | 80.0 | 40.0 | 26.7 | 46.7 | **38.5** (+1.5) |
| DFN ViT-H-14 | | $224^2$ | 986.1 | 20.0 | 26.7 | 73.3 | 26.7 | 26.7 | 66.7 | 46.7 | 13.3 | 53.3 | 39.3 |
| DFN ViT-H-14 | ✔ | $224^2$ | 986.1 | 20.0 | 20.0 | 80.0 | 40.0 | 46.7 | 66.7 | 46.7 | 20.0 | 53.3 | **43.7** (+4.4) |
| DFN ViT-H-14 | | $378^2$ | 986.7 | 13.3 | 20.0 | 53.3 | 33.3 | 26.7 | 66.7 | 40.0 | 20.0 | 40.0 | 34.8 |
| DFN ViT-H-14 | ✔ | $378^2$ | 986.7 | 26.7 | 26.7 | 53.3 | 33.3 | 26.7 | 73.3 | 26.7 | 13.3 | 60.0 | **37.8** (+3.0) |

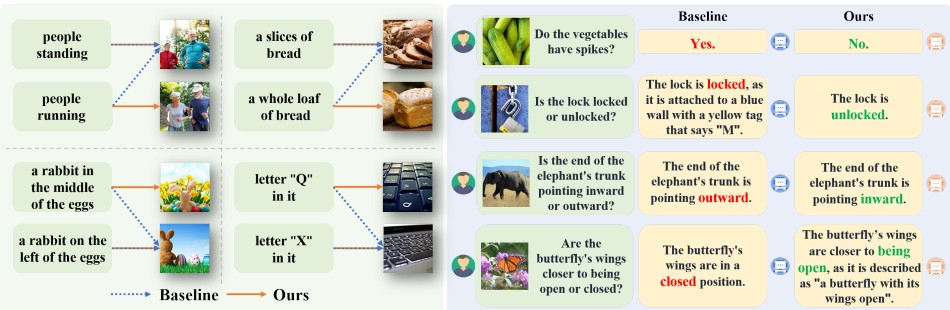

Figure 3: **Qualitative analysis on MMVP-VLM and MMVP benchmark. Left**: The prediction results from the OpenAI ViT-L-14 CLIP before & after incorporating **DIVA**. **Right**: The prediction results from LLaVA-1.5-7B before & after using our **DIVA**. The results on both benchmarks show that our framework can greatly enhance CLIP models' fine-grained visual perception capability and effectively alleviate the hallucination problem.

2023), MMBench (Liu et al., 2023), MMBench-CN (Liu et al., 2023), LLaVA-Bench-in-the-Wild (Liu et al., 2024b)). It can be clearly seen from Table 2 that LLaVA's performance is greatly boosted by replacing the original CLIP vision encoder to ours. The big accuracy gains on these benchmarks (except MME) are all thanks to the significant enhancement in CLIP's visual perception capabilities brought by our **DIVA** paradigm utilizing generative feedback.

Table 2: **Performance gains achieved by our enhanced CLIP visual backbone for MLLM (*i.e.*, LLaVA-1.5-7B and LLaVA-1.5-13B) on various V-L understanding tasks.** By refining the CLIP's representation with generative feedback, our method mitigates the visual deficiencies in MLLMs (*i.e.*, LLaVA$^{1.5}$) and improves original instruction following ability.

| Method | Ours | LLM | Image Size | MMVP | POPE rand | POPE pop | POPE adv | MME | MMBench en | MMBench cn | LLaVA-Wild |
|---|---|---|---|---|---|---|---|---|---|---|---|
| LLaVA$^{1.5}$ | | Vicuna-7B | $336^2$ | 24.7 | 87.3 | 86.1 | 84.2 | **1510.7** | 64.3 | 58.3 | 65.4 |
| LLaVA$^{1.5}$ | ✔ | Vicuna-7B | $336^2$ | **31.3** | 87.9 | **87.0** | 84.6 | 1500.6 | **66.4** | **60.6** | **66.3** |
| LLaVA$^{1.5}$ | | Vicuna-13B | $336^2$ | 30.7 | 87.1 | 86.2 | 84.5 | **1531.3** | 67.7 | **63.6** | 72.5 |
| LLaVA$^{1.5}$ | ✔ | Vicuna-13B | $336^2$ | **35.3** | **88.1** | **87.4** | 84.8 | 1522.9 | **69.4** | 63.1 | **73.5** |

Table 3: **Performance gains achieved by our enhanced CLIP backbone with generative guidance on semantic segmentation task.** * denotes the re-implemented results. Boosting CLIP's ability to perceive fine-grained visual details results in considerable benefits for visual dense prediction task.

| Method | Backbone | Ours | ADE20K-847 | ADE20K-150 | Pascal Context-459 | Pascal Context-59 |
|---|---|---|---|---|---|---|
| SAN* | ViT-L-14/224 | | 10.9 | 29.2 | 14.2 | 55.8 |
| SAN | ViT-L-14/224 | ✔ | 11.0 | **30.2** | **15.4** | **56.7** |
| SAN* | ViT-L-14/336 | | **11.5** | 30.3 | 14.7 | 56.7 |
| SAN | ViT-L-14/336 | ✔ | 11.5 | **31.8** | **15.7** | **57.8** |

**Enhanced Vision Backbone for Fine-Grained Visual Perception.** We also include segmentation task to evaluate the benefits brought by our enhanced CLIP backbones for visual dense prediction task. We adopt the recent state-of-the-art model in the open vocabulary semantic segmentation field, SAN (Xu et al., 2023c) with CLIP (Radford et al., 2021b) at both 224 and 336 image resolutions, as baselines. Four commonly used benchmarks (*i.e.*, ADE20K-847/150 (Zhou et al., 2017) and Pascal Context-459/59 (Mottaghi et al., 2014) are employed for performance evaluation. As shown in Table 3, with the benefit of our generative tuned CLIP backbone, the baseline models achieve considerable performance improvements on most of segmentation benchmarks and do not suffer performance degradation on the remaining one (*i.e.*, ADE20K-847).

## 4.4 GENERALIZATION CAPABILITY EVALUATION

Table 4: **Summary of zero-shot image classification performance on 27 datasets for evaluating model generalization capability.** O-1 and M-1 separately represent OpenAI ViT-L-14/224 and MetaCLIP ViT-H-14/224. **DIVA** greatly improves CLIP's ability to perceive visual details, while preserving its outstanding generalization capabilities.

| Method | Ours | ImageNet-1K | ImageNet-V2 | ImageNet-Adv. | ImageNet-Ren. | ImageNet-Ske. | ObjectNet | CIFAR-10 | CIFAR-100 | MNIST | Caltech-101 | SUN397 | FGVC Aircraft | Country-211 | Stanford Cars | Birdsnap | DTD | Eurosat | FER2013 | Flowers-102 | Food-101 | GTSRB | PCam | Pets | Rendered SST2 | Resisc45 | STL10 | VOC2007 | avg. top-1 acc. |
|---|---|---|---|---|---|---|---|---|---|---|---|---|---|---|---|---|---|---|---|---|---|---|---|---|---|---|---|---|---|
| O-1 | | 75.5 | 69.8 | 70.7 | 87.8 | 59.6 | 69.0 | 95.6 | 75.8 | 76.4 | 86.6 | 67.5 | 31.9 | 31.9 | 77.8 | 51.4 | 55.4 | 60.1 | 49.9 | 79.1 | 93.0 | 50.6 | 52.0 | 93.6 | 68.8 | 64.5 | 99.4 | 77.4 | 69.3 |
| O-1 | ✔ | 75.5 | 69.7 | 70.8 | 87.7 | 59.5 | 69.1 | 95.5 | 76.3 | 76.1 | 86.8 | 67.5 | 31.8 | 31.8 | 77.9 | 51.8 | 55.1 | 60.2 | 49.4 | 78.9 | 93.0 | 50.2 | 53.8 | 93.7 | 67.0 | 64.5 | 99.3 | 77.8 | 69.3 |
| M-1 | | 78.5 | 72.1 | 69.6 | 91.8 | 68.1 | 73.6 | 98.3 | 86.7 | 81.3 | 89.1 | 74.1 | 48.2 | 34.7 | 87.2 | 68.5 | 69.8 | 55.6 | 54.9 | 80.7 | 92.5 | 62.3 | 56.1 | 94.2 | 71.0 | 72.6 | 99.4 | 77.6 | 74.4 |
| M-1 | ✔ | 78.4 | 71.9 | 69.1 | 91.6 | 67.9 | 73.4 | 98.3 | 86.4 | 81.0 | 89.1 | 74.3 | 47.0 | 34.7 | 87.2 | 67.8 | 69.6 | 55.0 | 55.8 | 80.7 | 92.4 | 62.4 | 54.8 | 94.1 | 70.7 | 73.0 | 99.4 | 77.6 | 74.2 |

Table 5: **Summary of zero-shot image-to-text and text-to-image retrieval performance on Flickr30K (Young et al., 2014) and COCO (Lin et al., 2014) benchmark datasets for evaluating model generalization capability.** **DIVA** significantly enhances CLIP's visual detail perception ability while maintaining its excellent generalization capabilities.

| Method | Ours | Image Size | **Image-to-Text** Retrieval | | | | | | **Text-to-Image** Retrieval | | | | | |
|---|---|---|---|---|---|---|---|---|---|---|---|---|---|---|
| | | | Flickr30K | | | COCO | | | Flickr30K | | | COCO | | |
| | | | R@1 | R@5 | R@10 | R@1 | R@5 | R@10 | R@1 | R@5 | R@10 | R@1 | R@5 | R@10 |
| OpenAI ViT-L-14 | | 224² | 85.1 | 97.3 | 99.0 | 56.4 | 79.4 | 86.6 | 65.2 | 87.3 | 92.0 | 36.5 | 61.0 | 71.1 |
| OpenAI ViT-L-14 | ✔ | 224² | 85.3 | 97.3 | 99.0 | 56.7 | 79.7 | 87.0 | 64.4 | 86.9 | 92.0 | 36.6 | 61.0 | 71.3 |
| MetaCLIP ViT-H-14 | | 224² | 89.5 | 98.8 | 99.7 | 65.5 | 85.2 | 91.1 | 76.8 | 93.9 | 96.6 | 48.2 | 72.3 | 81.1 |
| MetaCLIP ViT-H-14 | ✔ | 224² | 89.2 | 98.7 | 99.7 | 65.5 | 85.0 | 91.1 | 77.3 | 93.8 | 96.7 | 48.4 | 72.4 | 81.1 |

After validation of **DIVA**'s ability to boost CLIP models' fine-grained visual perception abilities, we conduct a thorough assessment of CLIP model's original generalization ability. The details about all benchmarks can be found at Table 9 in Appendix. Specifically, OpenAI ViT-L-14 (Radford et al., 2021b) and MetaCLIP ViT-H-14 (Xu et al., 2023a), which are widely used and shows the greatest performance gains on MMVP-VLM benchmark, are adopted as our baselines. We present their zero-shot accuracies on 27 image classification benchmarks in Table 4. It is evident that our **DIVA** significantly enhances CLIP models' representations of fine-grained visual details without adversely affecting the generalization capabilities of the baselines to a large extent. Furthermore, Table 5 illustrates the comparison of zero-shot text-to-image and image-to-text retrieval performance before and after incorporating **DIVA**. The quantitative results reaffirm that optimizing CLIP models'

representations with **DIVA** preserves the original great generalization ability. Given that these tasks heavily rely on the CLIP visual backbone's global semantic understanding, it is reasonable that our generative guided CLIPs do not achieve much performance improvements on these tasks.

## 4.5 ABLATION STUDY

By taking OpenAI ViT-L-14/224 as the baseline model, we conduct comprehensive ablation studies on MMVP-VLM regarding each pattern and average score, exploring the effect of condition designing, introduced data scale and adopted diffusion models for **DIVA**.

Table 6: **Ablation study on the condition design for diffusion models.** G and L denote visual class token and patch tokens. Compared to using semantic matching constraints from image-text pairs as condition, taking visual features as condition for representation-level optimization is more effective. Furthermore, an appropriate degree of visual dense recap scheme is also crucial for **DIVA**.

| Visual Condition | Textual Condition | 🎨 | 🔍 | 🔁 | ↑↓ | 📍 | 🎨 | ⚙ | A | 📷 | Average |
|---|---|---|---|---|---|---|---|---|---|---|---|
| ✗ | ✗ | 13.3 | 13.3 | 20.0 | 20.0 | 13.3 | 53.3 | 20.0 | 6.7 | 13.3 | 19.3 |
| ✗ | Real Caption | 6.7 | 13.3 | 26.7 | 20.0 | 6.7 | 53.3 | 33.3 | 13.3 | 20.0 | 21.5 (+2.2) |
| ✗ | Free-Source Phrase | 6.7 | 13.3 | 20.0 | 20.0 | 6.7 | 53.3 | 26.7 | 13.3 | 26.7 | 20.7 (+1.4) |
| only G | Empty Caption | 20.0 | 20.0 | 20.0 | 20.0 | 13.3 | 46.7 | 26.7 | 20.0 | 13.3 | 22.2 (+2.9) |
| G + part_L | Empty Caption | 13.3 | 20.0 | 40.0 | 6.7 | 20.0 | 53.3 | 46.7 | 20.0 | 13.3 | **25.9** (+6.6) |
| G + all_L | Empty Caption | 6.7 | 20.0 | 40.0 | 6.7 | 6.7 | 40.0 | 40.0 | 6.7 | 13.3 | 20.0 (+0.7) |
| only G | Real Caption | 13.3 | 26.7 | 33.3 | 13.3 | 6.7 | 53.3 | 26.7 | 13.3 | 6.7 | 21.5 (+2.2) |
| G + part_L | Real Caption | 13.3 | 13.3 | 33.3 | 13.3 | 13.3 | 46.7 | 60.0 | 13.3 | 0.0 | 23.0 (+3.7) |
| G + all_L | Real Caption | 13.3 | 6.7 | 26.7 | 13.3 | 13.3 | 46.7 | 26.7 | 6.7 | 13.3 | 18.5 (-0.8) |

**Effect of Condition Design for Diffusion Models.** We first probe into the influence of diffusion models' condition designing. As elaborated in Sec. 3.3, the condition design of diffusion models is vital as it directly determines the upper limit of CLIP models' enhanced representation quality. As shown in Table 6, we consider two condition settings: 1) using pure text embeddings as condition, which similar to Diffusion-TTA (Prabhudesai et al., 2023) (rows 3-4); 2) incorporating densely recapped visual features and empty text's embeddings as condition (rows 5-7). For these two settings, the visual encoder is kept frozen for setting 1, while the text encoder stays frozen for setting 2. Specifically, whether using real caption matched with images from CC-3M dataset or using free-source phrases about image details, guiding CLIP through image-text matching constraints can yield performance gains. However, since this manner does not originate from the representation level, the achieved gain is not significant. In contrast, our condition design introduces appropriately densified visual features, constructing a framework that only uses images to achieve self-supervised optimization of CLIP representations and detaching from image-text form constraints. **DIVA** helps CLIP achieve the best performance gains by introducing partial visual local features coupled with class tokens as condition. Introducing too few or many local tokens results in visual density being too low (row 5) or high (row 7), both of which reduce the achieved performance improvement. We also explore whether incorporating text descriptions could further enhance **DIVA**'s potential. However, both our empirical intuitive and quantitative results indicate that the answer is negative. The main reason is that the introduction of text descriptions greatly reduces the difficulty of the reconstruction task, allowing the CLIP visual backbone's representations to easily accomplish the reconstruction without needing to optimize towards capturing more detailed representations. Therefore, it directly impairs the representation learning of the CLIP model's visual backbone through generative feedback.

Table 7: **Ablation study on the data scaling property of our DIVA with different data scales.** Training time is measured by # gpus×hours. **DIVA** demonstrates great potential with data scaling properties, where the increase in data volume proportionally enlarges the performance gains.

| Data Scale | Training Time | 🎨 | 🔍 | 🔁 | ↑↓ | 📍 | 🎨 | ⚙ | A | 📷 | Average |
|---|---|---|---|---|---|---|---|---|---|---|---|
| ✗ | ✗ | 13.3 | 13.3 | 20.0 | 20.0 | 13.3 | 53.3 | 20.0 | 6.7 | 13.3 | 19.3 |
| 25% | 16.8 | 6.7 | 13.3 | 20.0 | 13.3 | 6.7 | 46.7 | 53.3 | 13.3 | 13.3 | 20.7 (+1.4) |
| 50% | 32.8 | 13.3 | 13.3 | 40.0 | 13.3 | 6.7 | 26.7 | 53.3 | 20.0 | 13.3 | 22.2 (+2.9) |
| 75% | 49.6 | 6.7 | 26.7 | 40.0 | 13.3 | 6.7 | 53.3 | 53.3 | 6.7 | 6.7 | 23.7 (+4.4) |
| 100% | 66.4 | 13.3 | 20.0 | 40.0 | 6.7 | 20.0 | 53.3 | 46.7 | 20.0 | 13.3 | **25.9** (+6.6) |

**Data Scaling Property.** Then, we investigate the potential data scaling property of our **DIVA**. The corresponding results are presented in Table 7. It is obvious that the CLIP model's performance on MMVP-VLM benchmark is consistently improved with more training samples. As the ratios of introduced samples continue to rise, there's no sign of diminishing gains in accuracy, suggesting that our framework has great potential with continually scaled up training data. Noticeably, by integrating 100% training data, our method greatly boosts CLIP's visual perception capability (*i.e.*, approximately ↑7%) with 66.4 # gpus × hours. It means if 8 gpus are available, **DIVA** only need 8.3 hours training time to realize considerable performance gains for CLIP models. Besides, this setting results in an adaptation time of roughly 0.01 seconds per sample, proving the efficacy of our method.

Table 8: **Ablation study on the adopted generative diffusion models in our DIVA.** Training time is measured by # gpus×hours. Our framework is not sensitive to the version of stable diffusion models, consistently brings representation enhancement for CLIP models.

| Method | Diffusion Resolution | Training Time | 🧭 | 🔍 | 🔄 | ↑↕ | 💡 | 🌐 | ⚙️ | A | 📷 | Average |
|---|---|---|---|---|---|---|---|---|---|---|---|---|
| ✘ | ✘ | ✘ | 13.3 | 13.3 | 20.0 | 20.0 | 13.3 | 53.3 | 20.0 | 6.7 | 13.3 | 19.3 |
| DiT-XL/2 | $512^2$ | 80.8 | 20.0 | 13.3 | 20.0 | 6.7 | 6.7 | 20.0 | 20.0 | 6.7 | 6.7 | 13.3 (-6.0) |
| SD-1-4 | $512^2$ | 71.2 | 20.0 | 13.3 | 26.7 | 20.0 | 13.3 | 40.0 | 33.3 | 26.7 | 13.3 | 23.0 (+3.7) |
| SD-2-1-base | $512^2$ | 66.4 | 13.3 | 20.0 | 40.0 | 6.7 | 20.0 | 53.3 | 46.7 | 20.0 | 13.3 | **25.9** (+6.6) |
| SD-xl-base-1.0 | $512^2$ | 90.4 | 20.0 | 20.0 | 26.7 | 26.7 | 6.7 | 46.7 | 33.3 | 6.7 | 26.7 | 23.7 (+4.4) |

**Effect of Diffusion Model Structures.** At last, we explore the effect of incorporating various types of diffusion models for generative guidance. Specifically, two types of diffusion models are employed in our **DIVA**, including DiT (Peebles & Xie, 2023) and stable diffusion (SD) series (Rombach et al., 2022a). It is clear in Table 8 that our method achieves the biggest performance gain on MMVP-VLM by integrating SD-2-1-base. Furthermore, we observe that integrating DiT-XL/2 as generative guidance exacerbates the perceptual ability of original CLIP model in capturing visual details. We attribute this to DiT's relatively poor representation quality compared to SD models. For the included SD series, the quantitative results in Table 8 also demonstrate that **DIVA** is not sensitive to version of SD models, which can consistently refine CLIP's feature representations within our framework.

## 5 CONCLUSION

In this work, we focus on addressing the visual limitation of CLIP models that struggle with distinguishing fine-grained image details. We present the first work to explore leveraging generative feedback from text-to-image diffusion models to directly optimize CLIP models' representations. Specifically, by feeding dense visual features from CLIP as condition to the diffusion models and introducing the reconstruction loss from diffusion process onto the CLIP model's optimization, we establish a self-supervised framework **DIVA**. Notably, this architecture is simple and clean, requiring no additional plugins while demonstrating significant potential. Extensive evaluations demonstrate that our **DIVA** not only substantially enhances CLIP models' performance on the MMVP-VLM benchmark that measures visual abilities of vision-language models, but also aids in improving the performance of MLLMs and vision networks respectively on multimodal and visual understanding tasks. Furthermore, experiments on 29 benchmarks evaluating generalization capabilities confirm that our self-supervised **DIVA** maintains the CLIP models' original excellent generalization capabilities.

## 6 LIMITATIONS AND FUTURE TOPICS

One potential limitation of this work is that the data scale for generative fine-tuning and the model capacity of our **DIVA** could be scaled up further to push better CLIP representations and performance. Moreover, this work mainly focuses on designing a simple but effective framework to enhance CLIP models with generative diffusion process, which means although our **DIVA** demonstrates the newly exploited potential of using generative diffusion models for better CLIP models' representation guidance, it can be integrated with finer-grained supervision scheme to further boost discriminative model capabilities. Exploring additional modalities beyond image-text data, *e.g.*, video and audio, is also a promising direction for investigation. Furthermore, employing purely visual information when scaling up DIVA framework to further constrain the semantic alignment of CLIP's visual and text encoders is a worthwhile avenue for exploration. Since this work is just a beginning in this direction, it opens up a future research perspective to develop a more general and powerful framework based on diffusion models that could enhance vision-language foundation models.

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

APPENDIX

## A DATASETS FOR GENERALIZATION CAPABILITY EVALUATION

Table 9: **Datasets used to evaluate CLIP models's generalization capability**.

| Dataset | Classes | Test size | Evaluation Metric |
|---|---|---|---|
| ImageNet-1K (Deng et al., 2009) | 1000 | 50,000 | accuracy |
| ImageNet-V2 (Recht et al., 2019) | 1000 | 10,000 | accuracy |
| ImageNet-Adversarial (Hendrycks et al., 2021b) | 1000 | 7,500 | accuracy |
| ImageNet-R(endition) (Hendrycks et al., 2021a) | 1000 | 30,000 | accuracy |
| ImageNet-Sketch (Wang et al., 2019) | 1000 | 50,899 | accuracy |
| ObjectNet (Barbu et al., 2019) | 1000 | 50,273 | accuracy |
| CIFAR-10 (Krizhevsky et al., 2009) | 10 | 10,000 | accuracy |
| CIFAR-100 (Krizhevsky et al., 2009) | 100 | 10,000 | accuracy |
| MNIST (LeCun et al., 1998) | 10 | 10,000 | accuracy |
| Caltech101 (Fei-Fei et al., 2004) | 101 | 9144 | accuracy |
| SUN397 (Xiao et al., 2010) | 397 | 108,754 | accuracy |
| FGVC Aircraft (Maji et al., 2013) | 100 | 3,333 | accuracy |
| Country-211 (Radford et al., 2021a) | 211 | 21,100 | accuracy |
| Stanford Cars (Krause et al., 2013) | 196 | 8,041 | accuracy |
| Birdsnap (Berg et al., 2014) | 500 | 2,195 | accuracy |
| Describable Textures (Cimpoi et al., 2014) | 47 | 1,880 | accuracy |
| EuroSAT(Helber et al., 2019) | 10 | 27,000 | accuracy |
| Facial Emotion Recognition 2013 (Goodfellow et al., 2013) | 8 | 3,574 | accuracy |
| Oxford Flowers 102 (Nilsback & Zisserman, 2008) | 102 | 6,149 | accuracy |
| Food-101 (Bossard et al., 2014) | 102 | 25,250 | accuracy |
| GTSRB (Stallkamp et al., 2012) | 43 | 12,630 | accuracy |
| PatchCamelyon (Veeling et al., 2018) | 2 | 32,768 | accuracy |
| Oxford-IIIT Pets (Parkhi et al., 2012) | 37 | 3,669 | accuracy |
| Rendered SST2 (Radford et al., 2021a) | 2 | 1,821 | accuracy |
| RESISC45 (Cheng et al., 2017) | 45 | 31,500 | accuracy |
| STL-10 (Coates et al., 2011) | 10 | 8000 | accuracy |
| Pascal VOC 2007 Classification (Everingham et al., 2007) | 20 | 4,952 | accuracy |
| Flickr30K (Young et al., 2014) | - | 1000 | recall |
| COCO (Lin et al., 2014) | - | 5000 | recall |

## B PSEUDO CODE FOR **DIVA** PIPELINE

---
**Algorithm 1 `DIVA`**

---
1: **Input:** Image $x$, CLIP model weights $\theta$, diffusion model weights $\phi$, representation optimization steps $N$, batch size $B$, learning rate $\eta$, optimized CLIP model weights $\theta^*$.
2: **for** optimization step $s \in (1, \ldots, N)$ **do**
3:   Compute current CLIP's visual features $f\theta(x)$ as partial condition $\mathbf{c}$
4:   Sample timesteps $\{t_i\}_{i=1}^B$ and noises $\{\epsilon_i\}_{i=1}^B$
5:   Loss $L(\theta, \phi) = \frac{1}{N} \sum_{i=1}^B \|\epsilon_\phi(\sqrt{\bar{\alpha}_{t_i}}x + \sqrt{1 - \bar{\alpha}_{t_i}}\epsilon_i, \mathbf{c}, t_i) - \epsilon_i\|^2$
6:   Update CLIP weights $\theta^* \leftarrow \theta - \eta\nabla_\theta L(\theta, \phi)$
7: **end for**
8: **return** optimized CLIP weights $\theta^*$

---

## C DATASETS FOR FINE-GRAINED VISUAL PERCEPTION EVALUATION

The MMVP-VLM dataset (Tong et al., 2024b) evaluates CLIP models' sensitivity to fine-grained visual patterns. It includes nine distinct visual patterns, each with 15 image-text pairs. In a zero-shot setting, models must accurately match images to their corresponding textual descriptions. For each pattern, models align 15 image-text pairs from 30 images, with accuracy calculated as the

percentage of correct matches. A sample is correct only if the model accurately matches both the two images and texts. Overall performance is the average accuracy across all nine patterns, reflecting the model's understanding of various visual patterns. The MMVP dataset (Tong et al., 2024b) evaluates multimodal large language models' visual comprehension. It comprises 300 test images, each with associated questions and correct answers for visual question answering. Models are assessed in a zero-shot setting, generating answers based on image content and comparing them to predefined correct answers. Accuracy is calculated as the percentage of exact matches over the total questions. Overall performance is the average accuracy across all test images, reflecting the model's understanding of various visual patterns.

