# OpenReview forum: "Diffusion Feedback Helps CLIP See Better"
_ICLR.cc/2025/Conference — ICLR 2025 Poster_

### Official Review · Reviewer_UUiV · 2024-11-02

**Soundness:** 3
**Presentation:** 4
**Contribution:** 3
**Rating:** 8
**Confidence:** 5

**Summary:**

The paper introduces DIVA, a self-supervised framework designed to enhance CLIP's ability to perceive fine-grained visual details by leveraging generative feedback from text-to-image diffusion models. The framework incorporates a novel visual dense recap scheme, conditioning the diffusion models with CLIP's dense visual features and using image reconstruction loss for optimization. The results show that DIVA significantly improves CLIP's visual perception and its performance on multimodal and visual understanding tasks, as demonstrated on the MMVP-VLM benchmark. Additionally, DIVA maintains CLIP’s strong zero-shot capabilities across 29 image classification and retrieval benchmarks, proving its effectiveness in optimizing CLIP's discriminative representations.

**Strengths:**

--The paper presents an innovative approach by being the first to leverage generative feedback from text-to-image diffusion models to optimize the visual encoder of CLIP, aiming to addressing its limitation in fine-grained visual perception.

--The experiment part is extensive, validating their approach using the MMVP-VLM benchmark and demonstrating robust improvement. The integration with MLLM shows its impact, while experiments across 29 image classification and retrieval benchmarks confirm that the original capabilities of CLIP are preserved.

--The paper is clearly written, with well-explained methods and detailed descriptions of the proposed framework and experimental results.

**Weaknesses:**

--Disruption of Visual-Language Alignment: Optimizing CLIP’s visual encoder may disrupt the inherent visual-language alignment of CLIP. Although the authors validate that CLIP’s original capabilities are retained through experiments on 29 datasets, these tests are somewhat limited, as they only cover image classification and unimodal retrieval (text or image retrieval). Because the text encoder is not fine-tuned, the retrieval results for text remain unchanged, making text-retrieval experiments redundant to some degree. Moreover, the paper lacks experiments on tasks that rely heavily on cross-modal alignment, an area where CLIP excels, such as cross-modal retrieval. If such disruption occurs, it would be valuable to explore methods for preserving cross-modal alignment, such as applying a reconstruction loss to the visual class token to maintain the original feature.

--Part Token Selection Strategy: The method involves inputting tokens from CLIP’s visual encoder, including the global (class) token and a subset of local tokens, as conditions for the diffusion model generation. Although the authors state that using only the global token or a combination of the global token with all local tokens yields suboptimal results than global + part local tokens, they do not clarify how the proportion of local tokens is determined. Additionally, this proportion appears to vary across different datasets and between training and inference, raising concerns about the method’s generalizability. Furthermore, the strategy for selecting part tokens could impact the results, and using a simple random selection might not be optimal.

--Empty Text Conditions: In Table 6, the conditions for the diffusion model include an empty text condition. It would be interesting to explore whether incorporating text descriptions corresponding to the images could further enhance the results.

--Minor typo: There is an extra word "for" at line 350.

**Questions:**

1. It is crucial to also verify that the original cross-modal alignment capability of CLIP can also be preserved.

2. What criteria were used to decide the proportion of local tokens? Could you provide an analysis or ablation study to support this choice?

3. How does the varying proportion of local tokens across different datasets and between training and inference affect the method's generalizability?

4. Is there any other local token selection strategy that might yield better results?

5. Would incorporating image-associated text in the diffusion model conditions improve the performance observed in Table 6?

---

> ### Author Response · Authors · 2024-11-21
> **Response to Reviewer UUiV**
>
> Thanks very much for acknowledging the innovative research perspective, great potential of our DIVA framework, good writing and big efforts for comprehensive evaluation experiments. We have addressed all your questions in detail in the Response section below and have incorporated all feedback in the revision (*highlightened with blue color*). It is our sincere hope that our thorough explanations to the best of our abilities will contribute to an improved evaluation score of our work.
>
>
> **Our Responses to Paper Weaknesses:**
>
> > [**Q1**]. Disruption of Visual-Language Alignment: Optimizing CLIP's visual encoder may disrupt the inherent visual-language alignment of CLIP. Although the authors validate that CLIP's original capabilities are retained through experiments on 29 datasets, these tests are somewhat limited, as they only cover image classification and unimodal retrieval (text or image retrieval). Because the text encoder is not fine-tuned, the retrieval results for text remain unchanged, making text-retrieval experiments redundant to some degree. Moreover, the paper lacks experiments on tasks that rely heavily on cross-modal alignment, an area where CLIP excels, such as cross-modal retrieval. If such disruption occurs, it would be valuable to explore methods for preserving cross-modal alignment, such as applying a reconstruction loss to the visual class token to maintain the original feature. (It is crucial to also verify that the original cross-modal alignment capability of CLIP can also be preserved.)
>
>
> [**A1**]. Thank you very much for pointing out this valuable insight, but we believe there might be a significant misunderstanding that has led to the corresponding negative rating. Actually, we have thoroughly considered the issue of visual-language alignment during the development of this work. It is important to clarify that, beyond the experiments on 27 image classification datasets, which demonstrate that DIVA preserves the original CLIP model's strong cross-modal alignment, **the results reported in Table 5 specifically evaluate the cross-modal (image-to-text and text-to-image retrieval) performance of the CLIP model before and after DIVA generative fine-tuning. These are not, as you mentioned, experiments on "unimodal retrieval (text or image retrieval) tasks."** In light of your valuable suggestion, we have improved the relevant descriptions in the manuscript to minimize potential misunderstandings, with the updates highlighted in blue for clarity.

---

> ### Author Response · Authors · 2024-11-21
> **Response to Reviewer UUiV**
>
> > [**Q2**]. Part Token Selection Strategy: The method involves inputting tokens from CLIP's visual encoder, including the global (class) token and a subset of local tokens, as conditions for the diffusion model generation. Although the authors state that using only the global token or a combination of the global token with all local tokens yields suboptimal results than global + part local tokens, they do not clarify how the proportion of local tokens is determined. Additionally, this proportion appears to vary across different datasets and between training and inference, raising concerns about the method's generalizability. Furthermore, the strategy for selecting part tokens could impact the results, and using a simple random selection might not be optimal. (What criteria were used to decide the proportion of local tokens? Could you provide an analysis or ablation study to support this choice? How does the varying proportion of local tokens across different datasets and between training and inference affect the method's generalizability? Is there any other local token selection strategy that might yield better results?)
>
> [**A2**]. Thank you very much for this valuable comment and for your great interest in our work. Actually, our DIVA framework can actually improve fine-grained perception capabilities for all baselines through a simple and universal set of condition design principles. For instance, introducing all local patch tokens along with the class token as the visual condition for the diffusion model during training. The quantitative results of this approach are presented in Table 1 below.
>
> However, through experimental exploration, we find that the requirements for visual density vary across different baselines. **This is reasonable because the original training recipe for different CLIP models are fundamentally different (e.g., SigLIP utilizes sigmoid loss instead of contrastive loss), leading to variation in their learned representations.** Consequently, their visual density needs to be specifically adjusted to maximize the learning of fine-grained representations through generative feedback. The condition designs reported in our paper are derived from this insight. *In general, DIVA does not require finding the most optimal or designing complicated principle for visual density to enhance the fine-grained perception capabilities of CLIP models. However, tailoring the condition design to fit the characteristics of different baselines provides a more suitable setting and can further unleash the potential of DIVA.* **The rationale behind why condition designs differ for various CLIP models has been well explained in Sec. 3.3 to supplement the revised main text of our paper.**
>
> **Table 1: Performance of CLIP based models on various visual patterns of MMVP-VLM benchmark. Our DIVA can simultaneously improve CLIP's visual perception capability with a simple and universal condition designing scheme.**
> | Method | Image Size | Ours | Average |
> |:------|:-------:|:----:|:-----------:|
> | OpenAI ViT-L-14 | 224^2 |   | 19.3 |
> | OpenAI ViT-L-14 | 224^2 | √ | **20.0** |
> | OpenAI ViT-L-14 | 336^2 |   | 20.0 |
> | OpenAI ViT-L-14 | 336^2 | √ | **22.2** |
> | MetaCLIP ViT-L-14 | 224^2 |   | 23.7 |
> | MetaCLIP ViT-L-14 | 224^2 | √ | **27.4** |
> | MetaCLIP ViT-H-14 | 336^2 |   | 25.2 |
> | MetaCLIP ViT-H-14 | 336^2 | √ | **31.9** |
> | SigLIP ViT-SO-14 | 224^2 |   | 37.8 |
> | SigLIP ViT-SO-14 | 224^2 | √ | **39.3** |
> | SigLIP ViT-SO-14 | 384^2 |   | 37.0 |
> | SigLIP ViT-SO-14 | 384^2 | √ | **37.8** |
> | DFN ViT-H-14 | 224^2 |   | 39.3 |
> | DFN ViT-H-14 | 224^2 | √ | **43.7** |
> | DFN ViT-H-14 | 378^2 |   | 34.8 |
> | DFN ViT-H-14 | 378^2 | √ | **37.0** |

---

> ### Author Response · Authors · 2024-11-21
> **Response to Reviewer UUiV**
>
> > [**Q3**]. Empty Text Conditions: In Table 6, the conditions for the diffusion model include an empty text condition. It would be interesting to explore whether incorporating text descriptions corresponding to the images could further enhance the results. (Would incorporating image-associated text in the diffusion model conditions improve the performance observed in Table 6?)
>
> [**A3**]. Many thanks for your interests on our work and this comment. To further validate the rationale behind the condition design in our current DIVA framework, we have conducted additional experiments to examine whether incorporating real captions corresponding to the images can further enhance the performance benefits of the image-driven DIVA framework. However, both subjective insights and quantitative results (*as supplemented in Table 6 of our revised manuscript*) suggest that the answer is negative. The primary reason lies in the fact that introducing text descriptions corresponding to the images directly results in overly rich condition information. Given the provided text descriptions, the employed SD model can already reconstruct the original images easily. In other words, the inclusion of text descriptions substantially reduces the difficulty of the reconstruction task, thereby allowing the CLIP visual backbone to complete the reconstruction without needing to optimize towards more detailed representations. This directly impairs the ability of the CLIP model's visual backbone to improve its representations through generative feedback. As emphasized in Sec. 3.3 of our manuscript, "if the density is too high, the richness of the condition information approaches its maximum, greatly reducing the difficulty of the reconstruction task. This results in CLIP's representation requiring minimal optimization to easily complete the reconstruction, limiting the upper bound of CLIP's optimized capabilities". **We have supplemented the corresponding analysis into Sec. 4.5 of the revised main text.**
>
> Furthermore, the motivation behind DIVA is to establish an image-driven self-supervised learning approach that enhances the CLIP model's ability to perceive fine-grained details through a low-cost post-training process, rather than relying on image-text aligned data to improve the model. Introducing such data would misalign with our goals, as our DIVA framework is designed to offer notable advantages over directly fine-tuning the CLIP model with expensive image-text aligned data. Instead, DIVA demonstrates significant potential as a self-supervised representation learning method.
>
>
> > [**Q4**]. Minor typo: There is an extra word "for" at line 350.
>
> [**A4**]. Thanks for the kind reminder. We have corrected this typo per your valuable suggestion in our revised manuscript.

---

> > ### Comment · Reviewer_UUiV · 2024-11-26
> >
> > Thanks for the detailed response. The authors have addressed the primary concerns and questions comprehensively, providing detailed explanations and additional experiments to support the claims. Overall, I think this paper presents a reasonable and significant contribution to enhancing CLIP's fine-grained visual perception through an innovative self-supervised framework. Hence, I am inclined to provide a positive final recommendation.

---

> ### Author Response · Authors · 2024-11-27
> **Sincere Gratitude for Your Generous Recognition and Substantial Rating Improvement of Our Work**
>
> Thank you very much for updating your final score towards positive 8 with the valuable comment "this paper presents a reasonable and significant contribution to enhancing CLIP's fine-grained visual perception through an innovative self-supervised framework". **We sincerely appreciate your great recognition of our work and the efforts we have dedicated during the invaluable rebuttal period. We are truly gratified that our revised manuscript has addressed most of your concerns and convinced you of the solidity of our paper's content.** If you have any further questions, please do not hesitate to let us know and we will spare no effort to respond promptly.

---

### Official Review · Reviewer_gbrY · 2024-11-03

**Soundness:** 3
**Presentation:** 3
**Contribution:** 3
**Rating:** 8
**Confidence:** 3

**Summary:**

CLIP faces challenges related to fine-grained visual perception. To address this, the authors propose fine-tuning the CLIP image encoder by leveraging the spatial awareness of diffusion models. They use the visual tokens and the $[CLS]$ token from the CLIP image encoder as conditions for the diffusion models, backpropagating the gradients to update the CLIP encoder. The fine-tuned models are evaluated across several challenging benchmarks that require fine-grained visual perception, yielding promising results.

**Strengths:**

- Utilizing diffusion models to enhance CLIP's dense representation presents a novel perspective that will likely interest the community.

- The proposed method is intuitive and straightforward to implement, making it compatible with various existing vision-language models

- The experiments are comprehensive, demonstrating significant improvements across various benchmarks.

- The paper is well-written and easy to follow.

**Weaknesses:**

- Given that no explicit alignment constraint between the image and text encoders is applied during the fine-tuning process and only the image encoder is updated, I am concerned about the potential for misalignment between the image and text encoders.

- The visual recap operation varies across different models, potentially necessitating careful tuning of hyperparameters. Is there a general strategy applicable to all models, or must it be tailored to each specific model?

**Questions:**

- The proposed method significantly enhances the spatial awareness of CLIP. I wonder if this improvement extends to other multimodal tasks, such as open-vocabulary dense prediction tasks.

---

> ### Author Response · Authors · 2024-11-21
> **Response to Reviewer gbrY**
>
> Thanks very much for acknowledging the novel research perspective, great interest to the research community, great potential of our DIVA framework, good writing and big efforts for comprehensive evaluation experiments. We have addressed all your questions in detail in the Response section below and have incorporated all feedback in the revision (*highlightened with blue color*). It is our sincere hope that our thorough explanations to the best of our abilities will contribute to an improved evaluation score of our work.
>
>
> **Our Responses to Paper Weaknesses:**
>
> > [**Q1**]. Given that no explicit alignment constraint between the image and text encoders is applied during the fine-tuning process and only the image encoder is updated, I am concerned about the potential for misalignment between the image and text encoders.
>
> [**A1**]. Many thanks for you valuable comment. **Sincerely, this aspect has been carefully considered in our study, and the results presented in Tables 4 and 5 demonstrate that the alignment between CLIP's text encoder and visual encoder does not degrade.** In fact, the motivation behind DIVA is to employ an image-driven self-supervised learning approach to optimize CLIP's sensitivity to fine-grained details through a low-cost post-training process, rather than leveraging image-text aligned data to enhance its capabilities. *Compared to directly fine-tuning CLIP with image-text aligned data, our approach offers great advantages in this context and also provides intriguing insights for the research community.* Based on this motivation, it is natural that we do not incorporate textual annotations corresponding to the images as semantic alignment supervision for CLIP's image and text encoders.
>
> Our DIVA framework aims to refine CLIP's fine-grained visual representations with managable cost while maintaining its strong generalization ability. **Fundamentally, most post-training method inherently involves a trade-off with pre-training.** The vision-language alignment issue has been comprehensively addressed from the inception of this research, which is precisely what the experiments on 29 benchmark datasets in Tables 4 and 5 aim to demonstrate. Our results show that DIVA preserves the original strong generalization ability of the CLIP model while achieving substantial improvements in fine-grained representation capabilities, with only negligible sacrifices in generalization performance. This makes the DIVA framework highly promising. **Actually, employing purely visual information to further constrain the semantic alignment of CLIP's visual and text encoders is a worthwhile avenue for exploration. However, this is not the primary focus of our current work, which leaves further room for future exploration.**

---

> ### Author Response · Authors · 2024-11-21
> **Response to Reviewer gbrY**
>
> > [**Q2**]. The visual recap operation varies across different models, potentially necessitating careful tuning of hyperparameters. Is there a general strategy applicable to all models, or must it be tailored to each specific model?
>
> [**A2**]. Thank you very much for this valuable comment and for your great interest in our work. Actually, our DIVA framework can actually improve fine-grained perception capabilities for all baselines through a simple and universal set of condition design principles. For instance, introducing all local patch tokens along with the class token as the visual condition for the diffusion model during training. The quantitative results of this approach are presented in Table 1 below.
>
> However, through experimental exploration, we find that the requirements for visual density vary across different baselines. **This is reasonable because the original training recipe for different CLIP models are fundamentally different (e.g., SigLIP utilizes sigmoid loss instead of contrastive loss), leading to variation in their learned representations.** Consequently, their visual density needs to be specifically adjusted to maximize the learning of fine-grained representations through generative feedback. The condition designs reported in our paper are derived from this insight. *In general, DIVA does not require finding the most optimal or designing complicated principle for visual density to enhance the fine-grained perception capabilities of CLIP models. However, tailoring the condition design to fit the characteristics of different baselines provides a more suitable setting and can further unleash the potential of DIVA.* **The rationale behind why condition designs differ for various CLIP models has been well explained in Sec. 3.3 to supplement the revised main text of our paper.**
>
> **Table 1: Performance of CLIP based models on various visual patterns of MMVP-VLM benchmark. Our DIVA can simultaneously improve CLIP's visual perception capability with a simple and universal condition designing scheme.**
> | Method | Image Size | Ours | Average |
> |:------|:-------:|:----:|:-----------:|
> | OpenAI ViT-L-14 | 224^2 |   | 19.3 |
> | OpenAI ViT-L-14 | 224^2 | √ | **20.0** |
> | OpenAI ViT-L-14 | 336^2 |   | 20.0 |
> | OpenAI ViT-L-14 | 336^2 | √ | **22.2** |
> | MetaCLIP ViT-L-14 | 224^2 |   | 23.7 |
> | MetaCLIP ViT-L-14 | 224^2 | √ | **27.4** |
> | MetaCLIP ViT-H-14 | 336^2 |   | 25.2 |
> | MetaCLIP ViT-H-14 | 336^2 | √ | **31.9** |
> | SigLIP ViT-SO-14 | 224^2 |   | 37.8 |
> | SigLIP ViT-SO-14 | 224^2 | √ | **39.3** |
> | SigLIP ViT-SO-14 | 384^2 |   | 37.0 |
> | SigLIP ViT-SO-14 | 384^2 | √ | **37.8** |
> | DFN ViT-H-14 | 224^2 |   | 39.3 |
> | DFN ViT-H-14 | 224^2 | √ | **43.7** |
> | DFN ViT-H-14 | 378^2 |   | 34.8 |
> | DFN ViT-H-14 | 378^2 | √ | **37.0** |
>
>
> > [**Q3**]. The proposed method significantly enhances the spatial awareness of CLIP. I wonder if this improvement extends to other multimodal tasks, such as open-vocabulary dense prediction tasks.
>
> [**A3**]. Many thanks for your interest in our work and this comment. Based on your valuable suggestion, we have quickly supplemented experiments during the rebuttal period to evaluate whether the improvements made by our DIVA framework extend to other multimodal tasks (e.g., open-vocabulary dense prediction tasks), as shown in Table 2. Specifically, we include the previous state-of-the-art (SOTA) model for the visual grounding task (i.e., pixel-level referring expression segmentation), CRIS [1], as the baseline. Compared to the original CRIS model, replacing its original CLIP vision encoder with the one optimized by DIVA leads to consistent performance gains. This clearly demonstrates that while DIVA enhances the fine-grained representation capabilities of CLIP, it also provides continuous benefits for a wide range of vision-language understanding tasks. **Due to the time constraints of the rebuttal period, these performance gains could be further amplified. If time permits, we will update the results with even higher performance improvements.**
>
> **Table 2: Zero-shot and fine-tuned model performance gains achieved by our enhanced CLIP backbone with generative guidance on open-vocabulary dense prediction tasks (i.e., visual grounding). The mIoU is adopted as the standard evaluation metric.**
> | Method | Backbone | Ours | RefCOCO val | RefCOCO testA | RefCOCO testB | RefCOCO+ val | RefCOCO+ testA | RefCOCO+ testB | RefCOCOg val |
> |:------|:-------:|:----:|:-----------:|:-------------:|:-------------:|:------------:|:--------------:|:--------------:|:------------:|
> | CRIS | ViT-L-14/224 |   | 71.9 | 74.7 | 69.4 | 65.6 | 70.5 | 58.4 | 59.0 |
> | CRIS | ViT-L-14/224 | √ | **73.1** | **75.8** | **70.1** | **66.2** | **71.2** | **58.8** | **59.4** |
>
> [1] Wang Z, Lu Y, Li Q, et al. Cris: Clip-driven referring image segmentation[C]//Proceedings of the IEEE/CVF conference on computer vision and pattern recognition. 2022: 11686-11695.

---

> > ### Comment · Reviewer_gbrY · 2024-11-23
> >
> > Thanks for your detailed response , which addresses most of my concerns.  However, my concern regarding **[Q1] (alignment quality with language)** still remains, as noted by other reviewers. While the paper offers a novel and compelling insight into utilizing a visual-centric process to enhance CLIP's dense-level perception, it is crucial to examine the potential negative impacts on vision-language alignment quality in the absence of explicit alignment constraints.
> >
> > Although the results suggest that better dense-level understanding can be achieved without compromising such consistency, it remains unclear why gradient feedback from diffusion models does not negatively affect alignment with language embeddings. Furthermore, the paper lacks **a thorough analysis** of these dynamics, which could significantly enhance its contribution.  For instance, could the specific condition form employed during the training of the diffusion model or the model's network architecture influence alignment quality? Given that Tab. 8 shows feedback from different diffusion models does not consistently benefit CLIP models, further investigation in this area would be highly valuable.
> >
> > Nonetheless, I remain positive about the novelty and contributions of this paper and look forward to your further response.

---

> ### Author Response · Authors · 2024-11-23
> **Sincere Gratitude for Your Generous Recognition and Maintained Positive Rating of Our Work**
>
> **Thank you so much for maintaining your final score as 8 and remaining positive about the novelty and contributions of our paper. We sincerely appreciate your great recognition of our work and the efforts we have dedicated during the invaluable rebuttal period.** We are truly gratified that our revised manuscript has addressed most of your concerns, and have convinced you of the novel and compelling insight provided by our paper.
>
> We also thank you very much for your great interest in our work. We will as soon as possible endeavor to provide a corresponding analysis of the potential dynamics during the precious rebuttal. If you have any further questions, please do not hesitate to let us know and we will spare no effort to respond promptly.

---

### Official Review · Reviewer_hLU8 · 2024-11-04

**Soundness:** 2
**Presentation:** 1
**Contribution:** 3
**Rating:** 6
**Confidence:** 3

**Summary:**

The paper utilizes the diffusion model to improve CLIP features' spatial and visual understanding. As CLIPs are only trained with global visual token, it lacks the ability to understand fine-grained visual concepts. At the same time, the diffusion model is trained to model the image distribution, thus it inherently has a better visual understanding. Given this idea, the paper tries to fine-tune CLIP model with a diffusion model guidance.

In details, the proposed method sends the global token and random-sampled local tokens as a condition to a diffusion model. The gradient back from the diffusion loss is used as guidance to optimize the CLIP model. Based on this, the paper presents detailed ablation studies and shows improvement for visual understanding.

**Strengths:**

1. The paper follows a clear intuition on how to improve the CLIP model with better visual understanding. The utilization of diffusion model to improve CLIP model is overall a nice idea.

2. The paper presented a method to use both the global tokens and local tokens to improve the visual encoder. The paper also found that the percentage of used local tokens is important and conducted experiment to validate that.

2. The paper analyzes the method on multiple tasks and thus the improvement is better shown.

3. The paper shows ablation studies for key designs of the method (some are actually the analysis of the method).

**Weaknesses:**

Weaknesses:

- The paper's clarity of the method needs to be improved. Usually, the presentation (i.e., writing) of the paper would not be directly treated as weakness and some of them are put in suggestions below. However, the ambiguity in current method description does affect the reader's understanding of the paper method, and affect the assessment of the paper during review. Thus the reviewer put it as a weakness to highlight it and hope that it can be improved during rebuttal. I list some points here:

   1. For Sec 3.3, does it an expansion of context for the "visual features combined with the empty text’s embeddings"? If so, please help specify it in the text.
   2. From my current understanding, the visual features (patches) are feed as the text embeddings to the diffusion model pipeline. It would be better to have any analysis on the validity of doing this (at least have some discussions regarding this as it is not an natural operation).
   3. What would be the positional embedding for the patches when feeding into the text encoder?
   4. In Sec 3.3 ("visual reacp density"), it would be better to illustrate the reason why the random selection schemas are different for different methods.


- The paper lacks the details of the evaluation protocol. As the detailed inference method can largely affect the results, it's crucial to have a detailed description of how the method is evaluated on the MMVP dataset.
   1. Given this, it would be hard to evaluate the validity of the ablation studies, which also affect the review of the paper.

- The analysis needs to be improved in presentation:

   1. For some comparisons, the difference between baseline and proposed methods are close (e.g., 0.4 in absolute value). For these small differences, it's hard to understand its improvement. Some variance or significances analysis would be appreciated. Also, highlighting these numbers with gray colors would be helpful to readers.
   2. In Table 3, some baseline and the proposed methods have the same score, but only the proposed methods are bold. It would be confusing to readers.

Suggestions (will not be treated as paper weakness but would request it in the revision):
0. The algorithm description in the main text is not comprehensive enough for reproducing. It would be better to expand the descriptions. Even it sounds trivial, it would be better to extend these content for clarity.
1. It would be better to refine the terminology in Algo 1 (Appendix B). E.g.,
   - The $\theta^*$ and $\theta$ are confusing.
   - It's better to write CLIP feature as $f_\theta(x)$ instead of \theta(x)$

Comments:

I overall would like to trust the validness and soundness of the paper given the detailed experiments. The current details provided in the paper are not convincing enough to the readers.

**Questions:**

- The paper has highlighted that the CLIP is trained with limited length. Most diffusion models are also only trained with limited token lengths and have the same problem. Would it be possible that the golden fraction of local patch sampling rate (in Table 6) is just because of this?

---

> ### Author Response · Authors · 2024-11-21
> **Response to Reviewer hLU8**
>
> Thanks very much for acknowledging the nice research perspective, great potential of our DIVA framework and great efforts for comprehensive evaluation experiments. We have addressed all your questions in detail in the Response section below and have incorporated all feedback in the revision (*highlightened with blue color*). It is our sincere hope that our thorough explanations to the best of our abilities will contribute to an improved evaluation score of our work.
>
>
> **Our Responses to Paper Weaknesses:**
>
> > [**Q1**]. The paper's clarity of the method needs to be improved. Usually, the presentation (i.e., writing) of the paper would not be directly treated as weakness and some of them are put in suggestions below. However, the ambiguity in current method description does affect the reader's understanding of the paper method, and affect the assessment of the paper during review. Thus the reviewer put it as a weakness to highlight it and hope that it can be improved during rebuttal. I list some points here:
>
> 1. For Sec 3.3, does it an expansion of context for the "visual features combined with the empty text's embeddings"? If so, please help specify it in the text.
>
> 2. From my current understanding, the visual features (patches) are feed as the text embeddings to the diffusion model pipeline. It would be better to have any analysis on the validity of doing this (at least have some discussions regarding this as it is not an natural operation).
>
> 3. What would be the positional embedding for the patches when feeding into the text encoder?
>
> 4. In Sec 3.3 ("visual reacp density"), it would be better to illustrate the reason why the random selection schemas are different for different methods.
>
>
> [**A1**]. Thank you very much for your kind efforts to help us improve the quality of this work. We fully understand your point that "the presentation (i.e., writing) of the paper usually would not be directly treated as a weakness." Following your valuable suggestions, we have done our utmost to improve our manuscript during the rebuttal period and sincerely hope that the points you raised have been comprehensively addressed. We genuinely look forward to elevating your evaluation of this work. Next, we will address each of the points you mentioned in detail.
>
> 1. We understand the comment you raised here. To make it clearer and more precise for readers, we have updated the title of Sec. 3.3 to "Diffusion's Visual Condition Design". Additionally, beyond the explanation provided in Sec. 3.2, we have further emphasized in Sec. 3.3 that our condition is composed of the empty text embeddings from the diffusion model's text encoder and our densely recapped visual features.
>
> 2. Yes, your understanding is correct. The visual features are fed as text embeddings into the diffusion model pipeline, and we have carefully considered the rationale behind this approach. First, the motivation of our DIVA framework is to enhance CLIP's fine-grained representation capabilities through a resource-efficient post-training process in a purely image-driven self-supervised framework. Therefore, introducing real textual information as diffusion conditions is not aligned with our design principles. **However, bridging the gap between purely visual features and pure text embeddings is critical and has been carefully addressed.** After obtaining the visual features, we first feed them into a vision-language projection layer to map them into a high-dimensional representation space. Furthermore, to further narrow the gap between visual features and text embeddings, we incorporate the empty text embedding as part of the diffusion model condition. In this manner, our condition design can be thought of as a "visual sentence" wrapped in the shell of an empty text embedding, if this analogy helps clarify the concept. **This design ensures a more seamless integration with the diffusion model's condition input. We have added a detailed explanation of this rationale in Sec. 3.3 to provide further clarity.**
>
> 3. **We believe there might have been some misunderstanding. In fact, the image patches are not fed into the text encoder but are instead sent only to the CLIP visual encoder, whose representations need to be optimized.** The visual input image is first processed with positional embedding before the visual encoding in the CLIP vision encoder. These processed features are then combined with the empty text embedding to form the final condition for the diffusion model. *It is important to emphasize that our DIVA framework does not alter the original architecture of either the CLIP model or the diffusion model providing generative feedback.* The only aspect that requires specific design and adjustment is the condition design.

---

> ### Author Response · Authors · 2024-11-21
> **Response to Reviewer hLU8**
>
> 4. Actually, our DIVA framework can actually improve fine-grained perception capabilities for all baselines through a simple and universal set of condition design principles. For instance, introducing all local patch tokens along with the class token as the visual condition for the diffusion model during training. The quantitative results of this approach are presented in Table 1 below. However, through experimental exploration, we find that the requirements for visual density vary across different baselines. **This is reasonable because the original training recipe for different CLIP models are fundamentally different (e.g., SigLIP utilizes sigmoid loss instead of contrastive loss), leading to variation in their learned representations.** Consequently, their visual density needs to be specifically adjusted to maximize the learning of fine-grained representations through generative feedback. The condition designs reported in our paper are derived from this insight. *In general, DIVA does not require finding the most optimal or designing complicated principle for visual density to enhance the fine-grained perception capabilities of CLIP models. However, tailoring the condition design to fit the characteristics of different baselines provides a more suitable setting and can further unleash the potential of DIVA.* **The rationale behind why condition designs differ for various CLIP models has been well explained in Sec. 3.3 to supplement the revised main text of our paper.**
>
> **Table 1: Performance of CLIP based models on various visual patterns of MMVP-VLM benchmark. Our DIVA can simultaneously improve CLIP's visual perception capability with a simple and universal condition designing scheme.**
> | Method | Image Size | Ours | Average |
> |:------|:-------:|:----:|:-----------:|
> | OpenAI ViT-L-14 | 224^2 |   | 19.3 |
> | OpenAI ViT-L-14 | 224^2 | √ | **20.0** |
> | OpenAI ViT-L-14 | 336^2 |   | 20.0 |
> | OpenAI ViT-L-14 | 336^2 | √ | **22.2** |
> | MetaCLIP ViT-L-14 | 224^2 |   | 23.7 |
> | MetaCLIP ViT-L-14 | 224^2 | √ | **27.4** |
> | MetaCLIP ViT-H-14 | 336^2 |   | 25.2 |
> | MetaCLIP ViT-H-14 | 336^2 | √ | **31.9** |
> | SigLIP ViT-SO-14 | 224^2 |   | 37.8 |
> | SigLIP ViT-SO-14 | 224^2 | √ | **39.3** |
> | SigLIP ViT-SO-14 | 384^2 |   | 37.0 |
> | SigLIP ViT-SO-14 | 384^2 | √ | **37.8** |
> | DFN ViT-H-14 | 224^2 |   | 39.3 |
> | DFN ViT-H-14 | 224^2 | √ | **43.7** |
> | DFN ViT-H-14 | 378^2 |   | 34.8 |
> | DFN ViT-H-14 | 378^2 | √ | **37.0** |

---

> ### Author Response · Authors · 2024-11-21
> **Response to Reviewer hLU8**
>
> > [**Q2**]. The paper lacks the details of the evaluation protocol. As the detailed inference method can largely affect the results, it's crucial to have a detailed description of how the method is evaluated on the MMVP dataset. Given this, it would be hard to evaluate the validity of the ablation studies, which also affect the review of the paper.
>
> [**A2**]. **In fact, all the evaluation protocols used in our work strictly follow the evaluation standard outlined in [1], which introduces the MMVP and MMVP-VLM datasets.**
>
> Specifically, the MMVP-VLM dataset is designed to assess the ability of CLIP models to perceive fine-grained visual patterns. This dataset contains nine different visual patterns, each comprising 15 pairs of image-text pairs. The evaluation criterion requires the model to accurately match each pair of images and corresponding textual descriptions under zero-shot conditions. Specifically, for each visual pattern, the model must correctly match 15 pairs of image-text pairs from a pool of 30 images. Accuracy is measured as the percentage of correctly matched pairs out of the total number of pairs. Notably, a correct sample is only counted when the model accurately matches both an image pair and a text pair. The model's overall understanding and handling of various visual patterns are comprehensively evaluated by averaging its accuracy across all nine visual patterns.
>
> Additionally, the MMVP dataset is designed to evaluate the visual understanding capabilities of multimodal large language models. This dataset contains 300 test images, each paired with related questions and correct answers, specifically designed for the visual question-answering task. The evaluation criterion requires the model to answer the questions corresponding to each image accurately under zero-shot conditions. Specifically, the model generates answers based on the image content, which are then compared to the pre-defined correct answers. Accuracy is measured as the percentage of correct answers out of the total number of questions. Importantly, an answer is considered correct only if it exactly matches the correct answer. The model's overall ability to understand and process various visual patterns is assessed by averaging its accuracy across all test images.
>
> **Based on your precious suggestion, and due to the limited space in the main text, these detailed descriptions of the evaluation methods for the MMVP and MMVP-VLM datasets have been supplemented in the Appendix.**
>
> [1] Tong S, Liu Z, Zhai Y, et al. Eyes wide shut? exploring the visual shortcomings of multimodal llms[C]//Proceedings of the IEEE/CVF Conference on Computer Vision and Pattern Recognition. 2024: 9568-9578.
>
>
> > [**Q3**]. The analysis needs to be improved in presentation:
>
> 1. For some comparisons, the difference between baseline and proposed methods are close (e.g., 0.4 in absolute value). For these small differences, it's hard to understand its improvement. Some variance or significances analysis would be appreciated. Also, highlighting these numbers with gray colors would be helpful to readers.
>
> 2. In Table 3, some baseline and the proposed methods have the same score, but only the proposed methods are bold. It would be confusing to readers.
>
> [**A3**]. Many thanks for your interest in our work and for this comment.
>
> First, we believe the comparison you mentioned regarding "the difference between baseline and proposed methods being close" refers to the POPE benchmark results presented in Table 2. The evaluation settings in Table 2 are consistent with and fair under the standard evaluation protocol commonly adopted for vision-language models (e.g., LLaVA). *It is important to note that the POPE benchmark is designed to evaluate the hallucination of vision-language models.* **In contrast, our DIVA framework focuses on enhancing the CLIP model and MLLMs that use CLIP vision encoders as their visual backbone for fine-grained information perception.** *As a result, the improvements brought by DIVA on the POPE benchmark are not as significant as those observed on the MMVP or MMVP-VLM datasets.* However, this does not diminish the fact that DIVA significantly improves CLIP's performance on the challenging MMVP-VLM benchmark, which assesses fine-grained visual capabilities, and enhances the performance of MLLMs and vision models on multimodal understanding and segmentation tasks.
>
> Additionally, following your valuable suggestion, we have updated the manuscript to emphasize the few performance gains with absolute values lower than 1% in gray throughout the text. Regarding the issue you mentioned in Table 3, we have bolded the only identical score, "11.5", achieved by both the baseline and the proposed method to avoid any potential confusion for readers.

---

> ### Author Response · Authors · 2024-11-21
> **Response to Reviewer hLU8**
>
> > [**Q4**]. Suggestions (will not be treated as paper weakness but would request it in the revision):
> 1. The algorithm description in the main text is not comprehensive enough for reproducing. It would be better to expand the descriptions. Even it sounds trivial, it would be better to extend these content for clarity.
>
> 2. It would be better to refine the terminology in Algo 1 (Appendix B). E.g., The θ∗ and θ are confusing; it's better to write CLIP feature as fθ(x) instead of \theta(x)$.
>
>
> [**A4**]. Thank you very much for your valuable suggestion. To provide as much detail as possible to facilitate reproducibility, we have followed your advice and added more detailed descriptions of the algorithm in the main text. Specifically, we have included additional explanations for the equations in Sec. 3.1, expanded the descriptions in Sec. 3.2 regarding the sampling process over N diffusion steps, and added an explanation in Sec. 4.1 about the rationale behind the choice of the number of diffusion steps N. Additionally, we have further optimized the presentation of the pseudocode in the Appendix, with all relevant updates highlighted in blue for clarity.

---

> > ### Comment · Reviewer_hLU8 · 2024-11-21
> > **Updated score**
> >
> > Thanks for the detailed response and revisions. I think that the paper's content are solid and my primary concern before is missing technical details. As it has been largely resolved in revision, I have updated my score towards positive. -- Reviewer

---

> ### Author Response · Authors · 2024-11-21
> **Sincere Gratitude for Your Generous Recognition and Substantial Rating Improvement of Our Work**
>
> Thank you very much for updating your final score towards positive. We sincerely appreciate your great recognition of our work and the efforts we have dedicated during the invaluable rebuttal period. We are truly gratified that our revised manuscript has addressed all your concerns and convinced you of the solidity of our paper's content. If you have any further questions, please do not hesitate to let us know and we will spare no effort to respond promptly.

---

### Official Review · Reviewer_PoSC · 2024-11-04

**Soundness:** 2
**Presentation:** 2
**Contribution:** 2
**Rating:** 5
**Confidence:** 4

**Summary:**

This manuscript introduces DIVA, a fine-tuning framework to enhance CLIP's fine-grained visual capabilities using a text-to-image diffusion model. DIVA tunes CLIP through diffusion loss with a frozen diffusion model conditioned by the CLIP outputs. To be specific, DIVA introduces a heuristic design for CLIP output-conditioned diffusion model, which leverages randomly dropped CLIP output sequences. Throughout experiments, it demonstrates improvements in extensive benchmarks including MMVP-VLM.

**Strengths:**

- Learning a good discriminative representation from image generation loss is now new, but I think it still has a big potential and should be studied. In this regard, the proposed idea of fine-tuning a pre-trained discriminative model (CLIP) via a pre-trained generative model (Diffusion) is interesting and demonstrates its potential.

- The authors evaluate DIVA across various benchmarks, including multimodal understanding such as MMVP-VLM, the backbone of LLaVA, zero-shot classification, and segmentation.

**Weaknesses:**

- Overall, this manuscript lacks detailed motivations and explanations. It is unclear why this approach is needed - e.g., Why does CLIP need diffusion feedback and how does it help? Why this particular approach is superior to others?

- The training process involves sampling multiple random states of the diffusion process for each image, which can be computationally expensive. There is insufficient discussion on whether this cost is justified and whether the gains are significant enough under these costs.

- The proposed diffusion's condition design (section 3) is quite heuristic and overall improvements are not consistent and even not significant in many cases. For example, none of the cases in Tables 1, 2, 4, 5, 7, and 8 show consistent improvements in all metrics. Moreover, many metrics indicate the improvements are marginal so I am uncertain that the proposed method effectively works and can generalize well across different types of visual challenges..

- There is a lack of analysis on potential trade-offs when using diffusion models as feedback. The paper does not clearly address whether the alignment between the CLIP's text encoder and visual encoder might degrade over time, potentially affecting performance, despite there being no consideration of the text encoder in the proposed training framework.

**Questions:**

Please see the weakness section.

---

> ### Author Response · Authors · 2024-11-21
> **Response to Reviewer PoSC**
>
> We sincerely appreciate your friendliness and recognition of our interesting research perspective, great potential of our DIVA framework and great efforts for comprehensive evaluation experiments. We have addressed all of your questions in detail in the following Response section and have incorporated all feedback in the revision (*highlightened with blue color*). We hope that our detailed explanations will give us the precious opportunity to raise the evaluation score of our work in your perspective.
>
> **Our Responses to Paper Weaknesses:**
>
> > [**Q1**]. Overall, this manuscript lacks detailed motivations and explanations. It is unclear why this approach is needed - e.g., Why does CLIP need diffusion feedback and how does it help? Why this particular approach is superior to others?
>
> [**A1**]. Many thanks for this valuable comment. Please allow us to elaborate on the motivation of our work for better clarity. As explored in [1], the authors identify significant deficiencies in the visual perception capabilities of MLLMs, which stem from the limitations of the pretrained visual model, specifically the CLIP vision encoder. As shown in the failure cases illustrated in Fig. 1 and Fig. 3 of our manuscript, existing CLIP models often fail to differentiate the subtle visual differences between two similar images. Fundamentally, the inability of CLIP to distinguish fine-grained visual details arises from the contrastive learning paradigm and the noisy image-text pairs used during training. To address this issue, many recent MLLM designs (e.g., [1-3]) have explored ways to incorporate vision-only self-supervised models, such as DINOv2 [4], to enhance the visual perception capabilities of MLLMs. However, this approach does not fundamentally resolve the limitations of the CLIP vision encoder and introduces significant computational overhead.
>
> In contrast, our approach aims to integrate self-supervised learning scheme during training to fundamentally enhance the capabilities of the CLIP visual encoder, rather than relying on external visual feature encoders. In this context, *diffusion models, known for generating highly realistic and detailed images, inherently possess the fine-grained visual representation capabilities for text-to-image generation. To this end, we pioneeringly leverage diffusion models to provide feedback via text-to-image generation and construct a purely image-driven self-supervised learning approach to optimize the CLIP model's sensitivity to visual details with managable training costs.* **Our core superiority lies in enhancing CLIP's capabilities through resource-efficient post-training using only image data in a self-supervised manner. This approach fundamentally benefits from the detailed representation capabilities of diffusion models to boost CLIP, rather than relying on costly methods such as collecting high-quality datasets for fine-tuning or introducing additional visual encoders. The corresponding details have been thoroughly supplemented in the introduction section of our revised main text.**
>
> [1] Tong S, Liu Z, Zhai Y, et al. Eyes wide shut? exploring the visual shortcomings of multimodal llms[C]//Proceedings of the IEEE/CVF Conference on Computer Vision and Pattern Recognition. 2024: 9568-9578.
>
> [2] Kar O F, Tonioni A, Poklukar P, et al. BRAVE: Broadening the visual encoding of vision-language models[C]//European Conference on Computer Vision. Springer, Cham, 2025: 113-132.
>
> [3] Zong Z, Ma B, Shen D, et al. Mova: Adapting mixture of vision experts to multimodal context[J]. arXiv preprint arXiv:2404.13046, 2024.
>
> [4] Oquab M, Darcet T, Moutakanni T, et al. Dinov2: Learning robust visual features without supervision[J]. arXiv preprint arXiv:2304.07193, 2023.

---

> ### Author Response · Authors · 2024-11-21
> **Response to Reviewer PoSC**
>
> > [**Q2**]. The training process involves sampling multiple random states of the diffusion process for each image, which can be computationally expensive. There is insufficient discussion on whether this cost is justified and whether the gains are significant enough under these costs.
>
> [**A2**]. Thank you again for your comment. Regarding the diffusion sampling steps, our choice is made based on a comprehensive consideration of training cost and model performance gains. Specifically, we started from the most basic initial state, N=1, to evaluate its impact on improving the fine-grained representation quality of CLIP. As N increases, the training time cost also rises. When N is increased from 1 to 2 (i.e., each image undergoes diffusion sampling twice to provide two rounds of generative feedback for optimizing the CLIP model's representation), we observe performance improvements. However, further increasing N beyond 2 not only greatly escalates the training time cost but also fails to provide additional benefits for the representation learning of the CLIP model. Therefore, N=2 is identified as the "sweet spot" and is chosen as the final sampling step to uniformly improve the performance of various baselines.
>
> As emphasized in our ablation study on the data scaling property in Sec. 4.5 of the main text, setting N=2 as our final choice allows DIVA to enhance the fine-grained visual perception capabilities of OpenAI ViT-L-14 with only 66.4 GPU × hours. **This implies that, with 8 GPUs available, DIVA requires just 8.3 hours of training time to achieve considerable performance gains for CLIP models. Furthermore, this setting results in an adaptation time of approximately 0.01 seconds per sample**, which fully demonstrates the efficiency of our method. **Per your valuable suggestion, we have supplemented the corresponding discussion into Sec. 4.1 of the revised main text.**
>
>
> > [**Q3**]. The proposed diffusion's condition design (section 3) is quite heuristic and overall improvements are not consistent and even not significant in many cases. For example, none of the cases in Tables 1, 2, 4, 5, 7, and 8 show consistent improvements in all metrics. Moreover, many metrics indicate the improvements are marginal so I am uncertain that the proposed method effectively works and can generalize well across different types of visual challenges.
>
> [**A3**]. Thanks for this comment. We will now elaborate on the condition design and the performance improvements achieved by DIVA in detail.
>
> First, our DIVA framework can actually improve fine-grained perception capabilities for all baselines through a simple and universal set of condition design principles. For instance, introducing all local patch tokens along with the class token as the visual condition for the diffusion model during training. The quantitative results of this approach are presented in Table 1 below.
>
> However, through experimental exploration, we find that the requirements for visual density vary across different baselines. **This is reasonable because the original training recipe for different CLIP models are fundamentally different (e.g., SigLIP utilizes sigmoid loss instead of contrastive loss), leading to variation in their learned representations.** Consequently, their visual density needs to be specifically adjusted to maximize the learning of fine-grained representations through generative feedback. The condition designs reported in our paper are derived from this insight. *In general, DIVA does not require finding the most optimal or designing complicated principle for visual density to enhance the fine-grained perception capabilities of CLIP models. However, tailoring the condition design to fit the characteristics of different baselines provides a more suitable setting and can further unleash the potential of DIVA.* **The rationale behind why condition designs differ for various CLIP models has been well explained in Sec. 3.3 to supplement the revised main text of our paper.**
>
> **Table 1: Performance of CLIP based models on various visual patterns of MMVP-VLM benchmark. Our DIVA can simultaneously improve CLIP's visual perception capability with a simple and universal condition designing scheme.**
> | Method | Image Size | Ours | Average |
> |:------|:-------:|:----:|:-----------:|
> | OpenAI ViT-L-14 | 224^2 |   | 19.3 |
> | OpenAI ViT-L-14 | 224^2 | √ | **20.0** |
> | OpenAI ViT-L-14 | 336^2 |   | 20.0 |
> | OpenAI ViT-L-14 | 336^2 | √ | **22.2** |
> | MetaCLIP ViT-L-14 | 224^2 |   | 23.7 |
> | MetaCLIP ViT-L-14 | 224^2 | √ | **27.4** |
> | MetaCLIP ViT-H-14 | 336^2 |   | 25.2 |
> | MetaCLIP ViT-H-14 | 336^2 | √ | **31.9** |
> | SigLIP ViT-SO-14 | 224^2 |   | 37.8 |
> | SigLIP ViT-SO-14 | 224^2 | √ | **39.3** |
> | SigLIP ViT-SO-14 | 384^2 |   | 37.0 |
> | SigLIP ViT-SO-14 | 384^2 | √ | **37.8** |
> | DFN ViT-H-14 | 224^2 |   | 39.3 |
> | DFN ViT-H-14 | 224^2 | √ | **43.7** |
> | DFN ViT-H-14 | 378^2 |   | 34.8 |
> | DFN ViT-H-14 | 378^2 | √ | **37.0** |

---

> ### Author Response · Authors · 2024-11-21
> **Response to Reviewer PoSC**
>
> Regarding the performance improvements achieved by DIVA, let us carefully clarify all the experiments.
>
> 1. As demonstrated by the quantitative results in Tables 1, 2, and 3, DIVA significantly improves CLIP's performance on the challenging MMVP-VLM benchmark, which assesses fine-grained visual capabilities, with improvements ranging from 3% to 7%. Additionally, it enhances the performance of MLLMs and vision models on multimodal understanding and segmentation tasks. Although in a few metrics, DIVA does not yield performance gains (e.g., certain visual patterns in Table 1 or MME in Table 2), **this does not undermine the substantial performance improvements DIVA has achieved across the majority of metrics. These improvements have also been noted and acknowledged by other reviewers.**
>
> 2. As for your observation regarding the quantitative results in Tables 4 and 5 not showing consistent improvements across all metrics, these results exactly match our expectations. **The purpose of Tables 4 and 5 is to demonstrate that DIVA, as a resource-efficient post-training method, optimizes CLIP's fine-grained representation capabilities while maintaining its original strong generalization ability.**
>
> 3. In addition, Tables 6, 7, and 8 present our ablation studies, which aim to explore the impact of different condition designs, the data scaling property, and the effects of various diffusion models within the DIVA framework. The key takeaway from these tables is that the final DIVA setting allows CLIP models to achieve a 6.6% improvement on the MMVP benchmark. **The emphasis here is not on highlighting the performance gains brought by intermediate settings in the ablation studies but rather on showcasing the effectiveness of the final configuration.**
>
> If you believe additional experiments are necessary to further demonstrate the potential of DIVA, we will do our best to include relevant experiments during this precious discussion period.
>
>
> > [**Q4**]. There is a lack of analysis on potential trade-offs when using diffusion models as feedback. The paper does not clearly address whether the alignment between the CLIP's text encoder and visual encoder might degrade over time, potentially affecting performance, despite there being no consideration of the text encoder in the proposed training framework.
>
> [**A4**]. Many thanks for your comment. However, we respectfully disagree with the statement "There is a lack of analysis on potential trade-offs ... The paper does not clearly address whether the alignment ...". **This aspect has been carefully considered in our study, and the results presented in Tables 4 and 5 demonstrate that the alignment between CLIP's text encoder and visual encoder does not degrade.** In fact, the motivation behind DIVA is to employ an image-driven self-supervised learning approach to optimize CLIP's sensitivity to fine-grained details through a low-cost post-training process, rather than leveraging image-text aligned data to enhance its capabilities. *Compared to directly fine-tuning CLIP with image-text aligned data, our approach offers great advantages in this context and also provides intriguing insights for the research community.* Based on this motivation, it is natural that we do not incorporate textual annotations corresponding to the images as semantic alignment supervision for CLIP's image and text encoders.
>
> Our DIVA framework aims to refine CLIP's fine-grained visual representations with managable cost while maintaining its strong generalization ability. **Fundamentally, most post-training method inherently involves a trade-off with pre-training.** The vision-language alignment issue has been comprehensively addressed from the inception of this research, which is precisely what the experiments on 29 benchmark datasets in Tables 4 and 5 aim to demonstrate. Our results show that DIVA preserves the original strong generalization ability of the CLIP model while achieving substantial improvements in fine-grained representation capabilities, with only negligible sacrifices in generalization performance. This makes the DIVA framework highly promising. **Actually, employing purely visual information to further constrain the semantic alignment of CLIP's visual and text encoders is a worthwhile avenue for exploration. However, this is not the primary focus of our current work, which leaves further room for future exploration.**

---

> ### Comment · Reviewer_PoSC · 2024-11-27
>
> Thank you for your response. However, some of your replies still leave me uncertain. In particular, I remain unconvinced that the proposed methodology is effective in areas beyond MMVP. No new evidence has been provided to counter my point that the proposed method does not result in significant improvements across other domains. While I reviewed the newly provided Table 1, I believe it does not sufficiently address my concerns. Moreover, many metrics suggest that the improvements are marginal, leaving me doubtful about the proposed method's effectiveness and its generalizability across different types of visual challenges. Additionally, the following questions still remain:
>
> - Effectiveness Compared to Other Methods: After reading the author's response, I still do not understand why the proposed method is superior to other approaches. Even if a generative objective function is used to learn detailed visual representations, I would like to hear the author's insight into why the proposed method achieves better results.
>
> - Impact of Weak Noise and Sampling: The paper uses a very small number of samples (N=2) for computational efficiency, exposing the model to only very weak noise. This approach raises further questions. How is this different from merely analyzing clean images? Would the proposed method still perform effectively on strongly noisy samples (e.g., samples from the middle of the diffusion sequence)? The proposed approach appears to rely primarily on distillation from the diffusion model, benefiting from it rather than truly utilizing diffusion feedback.
>
> - Text Encoder and Training Limitations: I feel the response did not adequately address my concerns regarding the text encoder. Freezing the text encoder and independently training the image encoder seems to intuitively weaken the connection between the two. As I mentioned previously, prolonged training, or exposure to strong noise samples (e.g., large N or intermediate diffusion sequence samples), or more training data could cause the image encoder to change a lot, leading to a natural trade-off. While Table 4 and Table 5 show results from short training iterations (nearly 1epoch), they do not provide a sufficient basis for addressing this concern. If the limitations I raised are valid, then the proposed method may only remain effective under short training, which would impose a fundamental limitation on the methodology. It might be better to explicitly acknowledge this limitation or address it in another way.
>
> For these reasons, I will maintain my evaluation score.

---

> ### Author Response · Authors · 2024-11-27
> **Response to Reviewer PoSC**
>
> We are immensely grateful for your response. **We are relieved and pleased that our initial rebuttal and revised manuscript well addressed some of your concerns regarding our work, such as "insufficient discussion on whether this cost is justified and whether the gains are significant enough under these costs" and "why the proposed diffusion's condition design is quite heuristic".** However, as you have pointed out, some of our replies still leave you uncertain. **In fact, after carefully reviewing all of your comments on our work to date, we believe that the uncertainty stems from a lack of clarity in some details of our revised manuscript and rebuttal, leading to a significant misunderstanding of our work and, consequently, a negative rating.**
>
> Upon diligently examining our meticulously revised manuscript and our responses during the rebuttal period, most of other reviewers have either maintained their positive assessments of our work or significantly improved their final ratings, which indirectly demonstrates the strong recognition of our novel research perspective and the great potential of our DIVA framework. **To be honest, we highly value the precious time for discussion with each reviewer during the rebuttal stage. In the following, we will endeavor to address all points you have raised with as much clarity and brevity as possible. We hope that our detailed explanations will provide us with the invaluable opportunity to elevate the evaluation score of our work in your esteemed perspective.**
>
>
> **Our Responses to Paper Weaknesses:**
>
> > [**Q1**]. I remain unconvinced that the proposed methodology is effective in areas beyond MMVP. No new evidence has been provided to counter my point that the proposed method does not result in significant improvements across other domains. While I reviewed the newly provided Table 1, I believe it does not sufficiently address my concerns. Moreover, many metrics suggest that the improvements are marginal, leaving me doubtful about the proposed method's effectiveness and its generalizability across different types of visual challenges.
>
> [**A1**]. **We kindly ask you to understand that our DIVA, as a training-efficient post-training method, can only bring relatively limited performance gains compared to the high training costs of pre-training CLIP from scratch with lower-cost generative fine-tuning (e.g., for only 8 hours). What we are exploring in this work is how to bring the highest possible performance gains to CLIP with the lowest possible training costs. On this point, as a pioneering work in this novel insight direction, our proposed DIVA has already achieved this (the most direct evidence being that in Table 1 of the main text, on the MMVP-VLM dataset, DIVA has brought 3-7% performance gains on almost all baselines with lower training costs, which is certainly not negligible).** We ask you to pay attention to the performance gains achieved by our DIVA while also considering the actual training costs, and sincerely request that you take into account the positive evaluations and significant recognition from other reviewers regarding the considerable performance gains achieved by our designed DIVA across various V-L understanding tasks. **If you believe any additional experiments are necessary to further demonstrate the potential of DIVA, we will do our best to include relevant experiments during this precious discussion period.**
>
> 1. As demonstrated by the quantitative results in Tables 1, 2, and 3, the proposed DIVA as a training-efficient post-training method can significantly improve CLIP's performance on the challenging MMVP-VLM benchmark, which assesses fine-grained visual capabilities, **with improvements ranging from 3% to 7%**. Additionally, it enhances the performance of MLLMs and vision models on multimodal understanding and segmentation tasks. **Although on a few metrics, DIVA does not yield performance gains (e.g., certain visual patterns in Table 1 or MME in Table 2), this does not undermine the substantial performance improvements DIVA has achieved across the majority of metrics. These improvements have been noted and acknowledged by other reviewers.**

---

> ### Author Response · Authors · 2024-11-27
> **Response to Reviewer PoSC**
>
> 2. Furthermore, we kindly ask you to carefully review our related explanations. **In the first wave of our response during rebuttal, we did our utmost to newly supplement the experiments in Table 1 to demonstrate that our DIVA framework can actually improve fine-grained perception capabilities for all baselines through a simple and universal set of condition design principles**, but because the original training recipe for different CLIP models are fundamentally different (e.g., SigLIP utilizes sigmoid loss instead of contrastive loss), leading to variation in their learned representations. Consequently, their visual density needs to be specifically adjusted to maximize the learning of fine-grained representations through generative feedback. **The newly provided Table 1 is not intended to prove how much performance gain our DIVA can achieve with this unified set of condition design principles, and we have not made any overclaims accordingly.**
>
> 3. As for your observation regarding the quantitative results in Tables 4 and 5 not showing consistent improvements across all metrics, these results exactly match our expectations. **The purpose of Tables 4 and 5 is to demonstrate that DIVA, as a resource-efficient post-training method, optimizes CLIP's fine-grained representation capabilities while maintaining its original strong generalization ability.**
>
> 4. In addition, Tables 6, 7, and 8 present our ablation studies, which aim to explore the impact of different condition designs, the data scaling property, and the effects of various diffusion models within the DIVA framework. The key takeaway from these tables is that the final DIVA setting allows CLIP models to achieve a 6.6% improvement on the MMVP benchmark. **The emphasis here is not on highlighting the performance gains brought by intermediate settings in the ablation studies but rather on showcasing the effectiveness of the final configuration.**
>
>
> > [**Q2**]. Effectiveness Compared to Other Methods: After reading the author's response, I still do not understand why the proposed method is superior to other approaches. Even if a generative objective function is used to learn detailed visual representations, I would like to hear the author's insight into why the proposed method achieves better results.
>
> [**A2**]. We are deeply regretful that our initial response did not successfully convey to you the novelty and significant advantages of our proposed method. In this response, we aim to distill the most critical points to clarify our motivations and the superiority of our approach, hoping to elevate your assessment of our work.
>
> 1. **Research Background**: As explored in [1] recently, the authors firstly identify significant deficiencies in the visual perception capabilities of MLLMs, which fundamentally stem from the inability of CLIP to distinguish fine-grained visual details. To address this issue, many recent MLLM designs (e.g., [1-3]) have explored ways to incorporate self-supervised vision backbones, such as DINOv2 [4], to enhance the visual perception capabilities of MLLMs. **However, these approaches do not fundamentally resolve the limitations of the CLIP vision encoder and introduce significant computational overhead. In fact, previous works have largely overlooked the necessity of enhancing CLIP's fine-grained visual perception capabilities as pointed out in [1], which is why we are somewhat perplexed by the "other approaches" you mentioned in your comments, with only a few efforts such as [5] focusing on collecting costly high-quality multimodal data to directly fine-tune CLIP to unlock extended capabilities.**
>
> 2. **Our Insight**: Based on the aforementioned research background, our work focuses on how to efficiently and fundamentally guide CLIP to enhance its fine-grained visual perception capabilities under the constraints of a resource-efficient training recipe (i.e., **low training cost post-training**) and without the expense of collecting multimodal matching data (i.e., **self-supervised learning scheme**). With this direct motivation in mind, considering the current popularity of *diffusion models*, known for generating highly realistic and detailed images, *inherently possess the fine-grained visual representation capabilities for text-to-image generation*. **We aspire to explore how to utilize diffusion models as a direct visual assistant to teach CLIP to be more sensitive to visual fine-grained information.**

---

> ### Author Response · Authors · 2024-11-27
> **Response to Reviewer PoSC**
>
> 3. **Our Plan**: To this end, we pioneeringly leverage diffusion models to provide feedback via text-to-image generation and construct a purely image-driven self-supervised learning framework to optimize the CLIP model's sensitivity to visual details with low training costs. **Based on the prior that diffusion models with richer condition information can generate more ideal images, by inputting CLIP encoded visual features as conditions into the diffusion model, we expect that constraining the frozen diffusion model to act as a guidance bridge providing generative feedback for CLIP can generate images closer to the original, steering CLIP's visual representation from primarily high-level semantic features towards also possessing the detail features needed for image generation. This is the essence of why our designed DIVA framework can help CLIP model achieve better results on the MMVP-VLM dataset, which measures fine-grained visual perception capabilities.**
>
> 4. **Our Advantages**: Our core superiority lies in enhancing CLIP's capabilities through resource-efficient post-training using only image data in a self-supervised manner. **This approach fundamentally benefits from the detailed representation capabilities of diffusion models to boost CLIP, rather than relying on costly methods such as collecting high-quality datasets for fine-tuning or introducing additional visual encoders. The corresponding details have been thoroughly supplemented in the introduction section of our revised main text.**
>
> [1] Tong S, Liu Z, Zhai Y, et al. Eyes wide shut? exploring the visual shortcomings of multimodal llms[C]//Proceedings of the IEEE/CVF Conference on Computer Vision and Pattern Recognition. 2024: 9568-9578.
>
> [2] Kar O F, Tonioni A, Poklukar P, et al. BRAVE: Broadening the visual encoding of vision-language models[C]//European Conference on Computer Vision. Springer, Cham, 2025: 113-132.
>
> [3] Zong Z, Ma B, Shen D, et al. Mova: Adapting mixture of vision experts to multimodal context[J]. arXiv preprint arXiv:2404.13046, 2024.
>
> [4] Oquab M, Darcet T, Moutakanni T, et al. Dinov2: Learning robust visual features without supervision[J]. arXiv preprint arXiv:2304.07193, 2023.
>
> [5] Zhang B, Zhang P, Dong X, et al. Long-clip: Unlocking the long-text capability of clip[C]//European Conference on Computer Vision. Springer, Cham, 2025: 310-325.

---

> ### Author Response · Authors · 2024-11-27
> **Response to Reviewer PoSC**
>
> > [**Q3**]. Impact of Weak Noise and Sampling: The paper uses a very small number of samples (N=2) for computational efficiency, exposing the model to only very weak noise. This approach raises further questions. How is this different from merely analyzing clean images? Would the proposed method still perform effectively on strongly noisy samples (e.g., samples from the middle of the diffusion sequence)? The proposed approach appears to rely primarily on distillation from the diffusion model, benefiting from it rather than truly utilizing diffusion feedback.
>
> [**A3**]. **We believe there is a severe misunderstanding regarding the DIVA framework we proposed (especially the definition of the diffusion sampling step N in the diffusion model), which has led to your corresponding negative evaluation. In fact, the situation you claim, "exposing the model to only very weak noise," does not exist in our DIVA framework, and the corresponding questions and judgments are unreasonable.**
>
> As pinpointed in our revised manuscript, taking an **original image (i.e., without any added noise)** as input, the CLIP model encodes the corresponding visual features, which will be combined with the empty text's embeddings from the diffusion model's text encoder for the diffusion's condition. **Given the image with added noise, the diffusion model** attempts to predict the noise added from the previous step to the current step with the aforementioned condition. This process needs to be repeated N times because, for each image, we will randomly select N states with a uniform distribution from the total time steps (e.g., 0-1000) of the diffusion model for optimization. **In our work, N is chosen as 2, which means for each original image, the CLIP model learns to optimize the representation of visual fine-grained perception from the generative feedback provided by the diffusion denoising process at N=2 different randomly selected time steps, i.e., N represents the number of random selections of diffusion time steps (i.e., the number of random selections from 0-1000), which is why we emphasized in our first response to you that a larger N represents a greater training time cost. Your understanding is to directly treat N as the selected diffusion time step (i.e., a specific value within 0-1000), and you believe that a smaller N is closer to the early time step stage of diffusion, which adds only very weak noise to the original image. Your understanding is, in fact, incorrect.**
>
> **In summary, the inputs to the CLIP model and the diffusion model are the original clean image without noise and the noisy image with noise corresponding to the randomly selected diffusion time step, respectively. N determines how many diffusion time steps, which decide the noise state of the image, need to be randomly selected for each original image to provide generative feedback to the CLIP models. Therefore, your comment "the proposed approach appears to rely primarily on distillation from the diffusion model, benefiting from it rather than truly utilizing diffusion feedback" is actually unfair.** In contrast, focusing on overcoming CLIP's visual shortcomings in perceiving fine-grained details, *our proposed self-supervised framework DIVA is the first work to exploit the potential of leveraging generative feedback from text-to-image diffusion models to optimize CLIP model's discriminative representations.*

---

> ### Author Response · Authors · 2024-11-27
> **Response to Reviewer PoSC**
>
> > [**Q4**]. Text Encoder and Training Limitations: I feel the response did not adequately address my concerns regarding the text encoder. Freezing the text encoder and independently training the image encoder seems to intuitively weaken the connection between the two. As I mentioned previously, prolonged training, or exposure to strong noise samples (e.g., large N or intermediate diffusion sequence samples), or more training data could cause the image encoder to change a lot, leading to a natural trade-off. While Table 4 and Table 5 show results from short training iterations (nearly 1epoch), they do not provide a sufficient basis for addressing this concern. If the limitations I raised are valid, then the proposed method may only remain effective under short training, which would impose a fundamental limitation on the methodology. It might be better to explicitly acknowledge this limitation or address it in another way.
>
> [**A4**]. We deeply regret that our initial response did not adequately address your concerns regarding the text encoder, and we will now endeavor to elucidate our motivations with greater clarity.
>
> As we emphasized in our first wave of response, *the potential issue of text-image misalignment due to solely using generative feedback to reinforce CLIP's visual branch has been carefully considered in our early studies, and the results presented in Tables 4 and 5 demonstrate that the alignment between CLIP's text encoder and visual encoder does not degrade.* Fundamentally, most post-training methods inherently involve a trade-off with pre-training. **However, we sense that your current concern lies in that although our DIVA does not damage CLIP's originally excellent generalization ability and V-L alignment with the current training paradigm and the small amount of generative fine-tuning cost, you believe that longer training durations and more training data will inevitably cause CLIP's visual branch to gradually move away from the text encoder, leading to a semantic mismatch between the V-L encoders.**
>
> On this point, you are totally correct that without explicit text-image constraints, an excessive amount of generative tuning training (time or data) will inevitably lead to such issues. **However, this does not essentially align with the research perspective of our current work. Our work explores a diffusion feedback-based, low-cost post-training approach to optimize CLIP's sensitivity to fine-grained details. An excessive amount of generative tuning training has never been our intention in introducing the DIVA training recipe.** *As the first work to exploit the potential of leveraging generative feedback from text-to-image diffusion models to optimize CLIP model's representations*, **our current focus is on how to significantly enhance CLIP's visual perception capabilities with the smallest amount of data and training time possible.**
>
> **In fact, the issue of CLIP's visual branch gradually moving away from the text encoder with longer training durations and more training data is a potential problem we may consider when scaling up the DIVA framework in the future, but it in no way detracts from the potential that DIVA has already demonstrated under our current efficient and low-data training recipe, nor from the contribution of the insights we have proposed to the community.** We fully understand your high expectations and strict requirements for our work, but this is not something that can be fully researched and realized all at once; it requires more time and more researchers to devote their efforts. **Actually, for our plans to scale up the DIVA framework with more training data in the future, employing purely visual information to further constrain the semantic alignment of CLIP's visual and text encoders is a worthwhile avenue for exploration. However, this is not the primary focus of our current work, which leaves further room for future exploration. In light of your valuable feedback, we sincerely and explicitly acknowledge this potential limitation when scaling up the DIVA framework in future work, which has been clearly added into our "Limitations And Future Topics" section of the revised manuscript, highlighted in blue, and we hope that you will be satisfied with such supplement.**

---

> ### Author Response · Authors · 2024-12-04
> **To Reviewer PoSC：Respectful Reminder for Your Consideration of Our Wholehearted Response as the Discussion Deadline Approaches**
>
> Dear Reviewer PoSC,
>
> **It has been nearly a week since we first submitted our response to you, we would really love to touch base with you to see whether you had a chance to look at our wholehearted response.** We hope that it has helped address all the concerns you have raised in your reviews. **If there are other concerns or if you have more questions, we will be more than happy to provide additional clarification.**
>
> Thanks again for your valuable time! **We sincerely hope that you will find our work deserving of your esteemed recognition and that it may receive a promising final rating.**
>
> Best,
>
> Authors

---

### Official Review · Reviewer_Vv7J · 2024-11-11

**Soundness:** 3
**Presentation:** 3
**Contribution:** 3
**Rating:** 6
**Confidence:** 4

**Summary:**

The paper aims to improve some CLIP's shortcoming cases, e.g., understanding the quantity,color,structure of images, since CLIP is trained by focusing on the high level semantic features. Motivated by this issue, the paper proposed to leverage a diffusion model to assist the improvement of CLIPs and inject the CLIP features as a condition into the diffusion model. The design brought a significant performance gain on the MMVP-VLM benchmark as well as the Llava 1.5 and segmentation tasks without impairing the other benchmarks.

**Strengths:**

1. the paper is well written and easy to follow.
2. The improvement on the MMVP-VLM benchmark is significant.
3. The experiment is well designed and comprehensive.

**Weaknesses:**

1. The hypothesis of this paper is not well explained, and the solution is a little bit ad-hoc. In detail, the failure of CLIP on several typical cases mentioned in the paper should be better studied and analyzed instead of just citing a few papers and mentioning very shortly. Are the failure cases caused by the architecture design, training strategy or just the lack of training data is unknown. Only given such analysis, we can start to think of a solution to improve. On the contrary, the paper just proposed to use diffusion model to help CLIP output more detail or low-level features directly, which is less convincing to me.

2. Several settings of this paper are also ad-hoc. For instance, in L225 - L241, the paper discussed the 'Visual Recap Density'. It seems for different CLIP variants, the density is way different, and I can hardly find any relationship across them. A more scientific way to decide the density should be considered. Similar thing for the times of the condition is used. The paper chooses 2 (N=2). I'm curious how this number is determined and how the performance will change for different numbers, e.g., N=10, 100.

3. From my understanding, the paper used CC3M dataset to tune a pretrained CLIP. I wonder whether the gains on the benchmarks are just brought by tuning the CLIP with CC3M or the diffusion model?

4. The rows 3-4 in Table 6 is confusing. When the textual condition is used, how the vision encoder can be trained?

**Questions:**

See the weakness above

---

> ### Author Response · Authors · 2024-11-21
> **Response to Reviewer Vv7J**
>
> We sincerely appreciate your friendliness and recognition of our research perspective, good writing, great potential of our DIVA framework and the comprehensive experiments. We have addressed all of your questions in detail in the following Response section and have incorporated all feedback in the revision (*highlightened with blue color*). We genuinely hope that our detailed explanations and the additionally supplemented results with our best efforts will give us the precious opportunity to raise the evaluation score of our work in your perspective.
>
> **Our Responses to Paper Weaknesses:**
>
> > [**Q1**]. The hypothesis of this paper is not well explained, and the solution is a little bit ad-hoc. In detail, the failure of CLIP on several typical cases mentioned in the paper should be better studied and analyzed instead of just citing a few papers and mentioning very shortly. Are the failure cases caused by the architecture design, training strategy or just the lack of training data is unknown. Only given such analysis, we can start to think of a solution to improve. On the contrary, the paper just proposed to use diffusion model to help CLIP output more detail or low-level features directly, which is less convincing to me.
>
> [**A1**]. Many thanks for this valuable comment. Please allow us to elaborate on the motivation of our work for better clarity. As explored in [1], the authors identify significant deficiencies in the visual perception capabilities of MLLMs, which stem from the limitations of the pretrained visual model, specifically the CLIP vision encoder. As shown in the failure cases illustrated in Fig. 1 and Fig. 3 of our manuscript, existing CLIP models often fail to differentiate the subtle visual differences between two similar images. Fundamentally, the inability of CLIP to distinguish fine-grained visual details arises from the contrastive learning paradigm and the noisy image-text pairs used during training. To address this issue, many recent MLLM designs (e.g., [1-3]) have explored ways to incorporate vision-only self-supervised models, such as DINOv2 [4], to enhance the visual perception capabilities of MLLMs. However, this approach does not fundamentally resolve the limitations of the CLIP vision encoder and introduces significant computational overhead.
>
> In contrast, our approach aims to integrate self-supervised learning scheme during training to fundamentally enhance the capabilities of the CLIP visual encoder, rather than relying on external visual feature encoders. In this context, *diffusion models, known for generating highly realistic and detailed images, inherently possess the fine-grained visual representation capabilities for text-to-image generation. To this end, we pioneeringly leverage diffusion models to provide feedback via text-to-image generation and construct a purely image-driven self-supervised learning approach to optimize the CLIP model's sensitivity to visual details with managable training costs.* **Our core superiority lies in enhancing CLIP's capabilities through resource-efficient post-training using only image data in a self-supervised manner. This approach fundamentally benefits from the detailed representation capabilities of diffusion models to boost CLIP, rather than relying on costly methods such as collecting high-quality datasets for fine-tuning or introducing additional visual encoders. The corresponding details have been thoroughly supplemented in the introduction section of our revised main text.**
>
> [1] Tong S, Liu Z, Zhai Y, et al. Eyes wide shut? exploring the visual shortcomings of multimodal llms[C]//Proceedings of the IEEE/CVF Conference on Computer Vision and Pattern Recognition. 2024: 9568-9578.
>
> [2] Kar O F, Tonioni A, Poklukar P, et al. BRAVE: Broadening the visual encoding of vision-language models[C]//European Conference on Computer Vision. Springer, Cham, 2025: 113-132.
>
> [3] Zong Z, Ma B, Shen D, et al. Mova: Adapting mixture of vision experts to multimodal context[J]. arXiv preprint arXiv:2404.13046, 2024.
>
> [4] Oquab M, Darcet T, Moutakanni T, et al. Dinov2: Learning robust visual features without supervision[J]. arXiv preprint arXiv:2304.07193, 2023.

---

> ### Author Response · Authors · 2024-11-21
> **Response to Reviewer Vv7J**
>
> > [**Q2**]. Several settings of this paper are also ad-hoc. For instance, in L225 - L241, the paper discussed the 'Visual Recap Density'. It seems for different CLIP variants, the density is way different, and I can hardly find any relationship across them. A more scientific way to decide the density should be considered. Similar thing for the times of the condition is used. The paper chooses 2 (N=2). I'm curious how this number is determined and how the performance will change for different numbers, e.g., N=10, 100.
>
> [**A2**]. Thank you for this kind suggestion. We will elaborate on the condition design and diffusion sampling steps with more details in the following.
>
> First of all, our DIVA framework can actually improve fine-grained perception capabilities for all baselines through a simple and universal set of condition design principles. For instance, introducing all local patch tokens along with the class token as the visual condition for the diffusion model during training. The quantitative results of this approach are presented in Table 1 below.
>
> However, through experimental exploration, we find that the requirements for visual density vary across different baselines. **This is reasonable because the original training recipe for different CLIP models are fundamentally different (e.g., SigLIP utilizes sigmoid loss instead of contrastive loss), leading to variation in their learned representations.** Consequently, their visual density needs to be specifically adjusted to maximize the learning of fine-grained representations through generative feedback. The condition designs reported in our paper are derived from this insight. *In general, DIVA does not require finding the most optimal or designing complicated principle for visual density to enhance the fine-grained perception capabilities of CLIP models. However, tailoring the condition design to fit the characteristics of different baselines provides a more suitable setting and can further unleash the potential of DIVA.* **The rationale behind why condition designs differ for various CLIP models has been well explained in Sec. 3.3 to supplement the revised main text of our paper.**
>
> **Table 1: Performance of CLIP based models on various visual patterns of MMVP-VLM benchmark. Our DIVA can simultaneously improve CLIP's visual perception capability with a simple and universal condition designing scheme.**
> | Method | Image Size | Ours | Average |
> |:------|:-------:|:----:|:-----------:|
> | OpenAI ViT-L-14 | 224^2 |   | 19.3 |
> | OpenAI ViT-L-14 | 224^2 | √ | **20.0** |
> | OpenAI ViT-L-14 | 336^2 |   | 20.0 |
> | OpenAI ViT-L-14 | 336^2 | √ | **22.2** |
> | MetaCLIP ViT-L-14 | 224^2 |   | 23.7 |
> | MetaCLIP ViT-L-14 | 224^2 | √ | **27.4** |
> | MetaCLIP ViT-H-14 | 336^2 |   | 25.2 |
> | MetaCLIP ViT-H-14 | 336^2 | √ | **31.9** |
> | SigLIP ViT-SO-14 | 224^2 |   | 37.8 |
> | SigLIP ViT-SO-14 | 224^2 | √ | **39.3** |
> | SigLIP ViT-SO-14 | 384^2 |   | 37.0 |
> | SigLIP ViT-SO-14 | 384^2 | √ | **37.8** |
> | DFN ViT-H-14 | 224^2 |   | 39.3 |
> | DFN ViT-H-14 | 224^2 | √ | **43.7** |
> | DFN ViT-H-14 | 378^2 |   | 34.8 |
> | DFN ViT-H-14 | 378^2 | √ | **37.0** |
>
> Regarding the diffusion sampling steps, our choice is made based on a comprehensive consideration of training cost and model performance gains. Specifically, we started from the most basic initial state, N=1, to evaluate its impact on improving the fine-grained representation quality of CLIP. As N increases, the training time cost also rises. When N is increased from 1 to 2 (i.e., each image undergoes diffusion sampling twice to provide two rounds of generative feedback for optimizing the CLIP model's representation), we observe performance improvements. However, further increasing N beyond 2 not only greatly escalates the training time cost but also fails to provide additional benefits for the representation learning of the CLIP model. Therefore, N=2 is identified as the "sweet spot" and is chosen as the final sampling step to uniformly improve the performance of various baselines. **Per your valuable suggestion, we have supplemented the corresponding discussion into Sec. 4.1 of the revised main text.**

---

> ### Author Response · Authors · 2024-11-21
> **Response to Reviewer Vv7J**
>
> > [**Q3**]. From my understanding, the paper used CC3M dataset to tune a pretrained CLIP. I wonder whether the gains on the benchmarks are just brought by tuning the CLIP with CC3M or the diffusion model?
>
> [**A3**]. Thank you for raising this valuable comment. In fact, we have considered this critical point in the early exploration of this work, minimizing direct benefits from data and instead focusing on exploring the potential of the DIVA framework itself to optimize CLIP's representations. The choice of the CC-3M dataset is essentially motivated by its relatively high-quality image data.
>
> Based on your valuable suggestion, to intuitively verify that the gains on the included benchmarks are essentially brought by tuning CLIP with our DIVA framework rather than leveraging the unseen CC-3M data, we have further supplement experiments using CLIP models pre-trained on the CC-3M and CC-12M datasets as baselines. **We aim to demonstrate that DIVA can enhance CLIP's fine-grained visual perception capability without requiring new training data that the CLIP model has not seen during its pre-training phase.** The employed baseline models are provided by this repository "https://github.com/facebookresearch/SLIP". As shown in Table 2 below, our DIVA framework significantly enhances the performance of baseline CLIP model (i.e., ViT-B-16/224) on the MMVP benchmark, which evaluates fine-grained visual perception. Moreover, we also include the CLIP model originally pre-trained on the larger-scale CC-12M dataset as an additional baseline to further validate the potential of DIVA. As shown in Table 2, even for the CLIP model pre-trained on the larger CC-12M dataset, our DIVA framework achieves considerable performance improvements on MMVP.**Due to the time constraints of the rebuttal period, these performance gains could be further amplified. If time permits, we will update the results with even higher performance improvements in the near future.**
>
> **Table 2: Performance of CLIP based models on various visual patterns of MMVP-VLM benchmark. Without introducing any additional unseen data to post-train the CLIP model, our DIVA greatly boost fine-grained visual perception capability.**
> | Method | Pretraining Data | Ours | Average |
> |:------|:-------:|:----:|:-----------:|
> | ViT-B-16/224 | CC-3M |   | 8.9 |
> | ViT-B-16/224 | CC-3M | √ | **11.9** |
> | ViT-B-16/224 | CC-12M |   | 8.1 |
> | ViT-B-16/224 | CC-12M | √ | **10.4** |
>
>
> > [**Q4**]. The rows 3-4 in Table 6 is confusing. When the textual condition is used, how the vision encoder can be trained?
>
> [**A4**]. Thanks for this comment, and apologies for the confusion. To clarify, when we introduce text information as the sole condition for the Stable Diffusion (SD) model (as shown in rows 3-4 of Table 6), the visual encoder of the CLIP model does not participate in model weight updates. Here, we aim to demonstrate that updating only the text encoder yields some benefits but falls significantly short of the improvements brought by DIVA in enhancing the corresponding capabilities of the CLIP visual encoder. **We have added further clarifications to the relevant Sec. 4.5 of the main text to prevent potential misunderstandings.**
>
> Additionally, to further validate the rationale behind the condition design in our current DIVA framework, we have conducted additional experiments to examine whether incorporating real captions corresponding to the images can further enhance the performance benefits of the image-driven DIVA framework. However, both subjective insights and quantitative results (*as supplemented in Table 6 of our revised manuscript*) suggest that the answer is negative. The primary reason lies in the fact that introducing text descriptions corresponding to the images directly results in overly rich condition information. Given the provided text descriptions, the employed SD model can already reconstruct the original images easily. In other words, the inclusion of text descriptions substantially reduces the difficulty of the reconstruction task, thereby allowing the CLIP visual backbone to complete the reconstruction without needing to optimize towards more detailed representations. This directly impairs the ability of the CLIP model's visual backbone to improve its representations through generative feedback. **We have supplemented the corresponding analysis into Sec. 4.5 of the revised main text.**
>
> Besides, the motivation behind DIVA is to establish an image-driven self-supervised learning method that enhances the CLIP model's ability to perceive fine-grained details through a low-cost post-training process, rather than relying on image-text aligned data to improve the model. Introducing such data would misalign with our goals, as our DIVA framework is designed to offer notable advantages over directly fine-tuning the CLIP model with expensive image-text aligned data. Instead, DIVA demonstrates significant potential as a self-supervised representation learning method.

---

> ### Author Response · Authors · 2024-12-04
> **To Reviewer Vv7J：Respectful Reminder for Your Consideration of Our Wholehearted Response as the Discussion Deadline Approaches**
>
> Dear Reviewer Vv7J,
>
> **It has been nearly two weeks since we first submitted our response to you, we would really love to touch base with you to see whether you had a chance to look at our wholehearted response.** We hope that it has helped address all the concerns you have raised in your reviews. **If there are other concerns or if you have more questions, we will be more than happy to provide additional clarification.**
>
> Thanks again for your valuable time! **We sincerely hope that you will find our work deserving of your esteemed recognition and that it may receive a promising final rating.**
>
> Best,
>
> Authors

---

### Author Response · Authors · 2024-11-21
**Overall Response**

We thank reviewers for all the valuable feedback, and the positive comments on **meaningful research perspective** (*Reviewer Vv7J*, *Reviewer PoSC*, *Reviewer hLU8*, *Reviewer gbrY*, *Reviewer UUiV*), **potential contributions to the community** (*Reviewer Vv7J*, *Reviewer PoSC*, *Reviewer hLU8*, *Reviewer gbrY*, *Reviewer UUiV*), **good writing** (*Reviewer Vv7J*, *Reviewer gbrY*, *Reviewer UUiV*) and **extensive evaluations and comparisons** (*Reviewer Vv7J*, *Reviewer PoSC*, *Reviewer hLU8*, *Reviewer gbrY*, *Reviewer UUiV*).

We address all the reviewers' comments below and have incorporated all feedback in the revised manuscript (*highlightened with blue color*). **We sincerely aspire that our detailed rebuttal will dispel any uncertainties or misunderstandings which reviewers may have raised regarding our manuscript, thus contributing positively to the final ratings of this work. If any additional experiments are needed to further demonstrate the potential of DIVA, we will do our utmost to supplement the relevant experiments during the valuable discussion period.**

---

### Author Response · Authors · 2024-11-30
**Sincere Request Again for Possible Review of Our Wholehearted Detailed Response**

Dear Reviewer Vv7J and Reviewer PoSC,

We would like to touch base with you to see whether you had a chance to look at our wholehearted response. We hope that it has helped address all the concerns you have raised in your reviews. **If there are other concerns or if you have more questions, we will be more than happy to provide additional clarification.**

Thanks again for your valuable time! **We sincerely hope that you will find our work deserving of your esteemed recognition and that it may receive a promising final rating.**

Best,

Authors

---

### Meta-Review · Area_Chair_YgDJ · 2024-12-18

**Metareview:**

This paper introduces DIVA, a post-training technique for CLIP-like models to overcome some of their shortcomings using a self-supervised diffusion approach that is trained from images only, using the CLIP feature to condition the diffusion model. This approach improves performance on fine-grained visual recognition, multimodal understanding and segmentation tasks.
The reviews mention several strengths, including that the paper is well written, significant improvements over prior work, well executed experiments, interesting novel direction, multiple tasks are considered, method compatible with various existing VLMs.
Weaknesses: effect of finetuning on CC3M not ablated, lack of detail on the cost of the sampling process, lack of clarity in places (method description and evaluation protocol).

**Additional Comments On Reviewer Discussion:**

In response to the reviews the authors submitted a detailed rebuttal and revised version of the manuscript. The rebuttal successfully addressed most points raised in the reviews, and four out of five reviewers recommend accepting the paper after the rebuttal. The remaining reviewer rates the paper as marginally below the acceptance threshold, despite the extensive clarifications in the author response. The AC does not see major concerns preventing to follow the majority reccommedation to accept the paper.

---

### Decision · Program_Chairs · 2025-01-22

Accept (Poster)